# FP8 Quantization: The Power of the Exponent

**Andrey Kuzmin**[*], **Mart Van Baalen**[*], **Yuwei Ren**,
**Markus Nagel, Jorn Peters, Tijmen Blankevoort**
Qualcomm AI Research[†]
{akuzmin,mart,ren,markusn,jpeters,tijmen}@qti.qualcomm.com

## Abstract

When quantizing neural networks for efficient inference, low-bit integers are the go-to format for efficiency. However, low-bit floating point numbers have an extra degree of freedom, assigning some bits to work on an exponential scale instead. This paper in-depth investigates this benefit of the floating point format for neural network inference. We detail the choices that can be made for the FP8 format, including the important choice of the number of bits for the mantissa and exponent, and show analytically in which settings these choices give better performance. Then we show how these findings translate to real networks, provide an efficient implementation for FP8 simulation, and a new algorithm that enables the learning of both the scale parameters and number of exponent bits in the FP8 format. Our chief conclusion is that when doing post-training quantization for a wide range of networks, the FP8 format is better than INT8 in terms of accuracy, and the choice of the number of exponent bits is driven by the severity of outliers in the network. We also conduct experiments with quantization-aware training where the difference in formats disappears as the network is trained to reduce the effect of outliers[1].

## 1 Introduction

Neural network quantization is one of the most effective ways to improve the efficiency of neural networks. Quantization allows weights and activations to be represented in low bit-width formats, e.g. 8 bit integers (INT8). When executing networks on any device, this leads to a reduction in data movement and enables the use of low bit-width computations, resulting in significantly faster inference and lower energy consumption.

Roughly speaking, values in neural networks are represented in either integer (INT) or floating-point formats (FP). Most quantization research has gone into low-bit integer formats such as INT8 and INT4, as the corresponding hardware for this is widely available. However, some research suggests there might be benefits to using low bit-width floating-point formats for neural network computations [40].

In this paper, we set out to thoroughly investigate the potential benefits of the floating point format for neural network inference. We show that floating point numbers add an extra dimension on top of the INT8 format that makes outliers in the quantized distribution less harmful to the overall performance. We investigate the effect of the quantization formats on neural network quantization on three levels: 1) Analytically for several common data and weight distributions, 2) practically in INT8 and FP8 post-training quantization (PTQ) settings, and 3) in quantization-aware training (QAT) settings with both INT8 and different FP8 formats. We will show there is a strong agreement between our theoretical results and our practical results on real networks.

---

[*]Equal contribution

[†]Qualcomm AI Research is an initiative of Qualcomm Technologies, Inc.

[1]Code will be made available at `https://github.com/Qualcomm-AI-research/FP8-quantization`

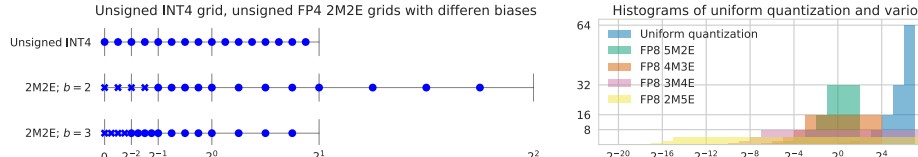

Figure 1: **Left:** Unsigned INT4 quantization compared to unsigned FP4 2M2E quantization with two different biases. Subnormal ranges are marked with crosses, normal range with circles. Compared to uniform quantization, FP quantization has larger dynamic range, but less precision for larger values. **Right**: Histograms of the positive values representable by 8 bit signed integers and signed 8 bit floating point formats. Histogram bins are log-spaced, to show the relation between dynamic range (i.e. number of bins) and precision (i.e. values per bin) between different formats. FP8 formats allow values with larger dynamic range, at the expense of precision. Intuitively, changing the (floating point) bias value for each distribution would move these histograms to the left or right without changing their shape.

In order to compare the two formats, we introduce a novel floating-point quantization simulation implementation that enables us to quickly run PTQ and QAT experiments with many floating point formats. Furthermore, this quantizer enables us to learn the FP8 exponent bias value, as well as the best trade-off between the number of bits used for the exponent and the mantissa, making good use of the flexibility the floating-point format provides and providing a way to learn this automatically without manual intervention. Our conclusion from the study presented in this paper is that floating-point quantization can be better than integer quantization for inference, but it needs a carefully tuned bias term and correct bit-settings between the exponent and mantissa.

## 2 Background

### 2.1 Integer quantization

Quantization is often applied to enable efficient integer matrix multiplications instead of expensive 32-bit floating point (FP32) multiplications. A matrix $\boldsymbol{X} \in \mathbb{R}^{m \times n}$ is quantized to an integer matrix $\boldsymbol{X}^{(int)}$ with an associated scale $s$ as follows: $\boldsymbol{X}^{(int)} = \text{clip}\left(\left\lfloor\frac{\boldsymbol{X}}{s}\right\rceil, x_{min}, x_{max}\right)$, where $\lfloor\cdot\rceil$ indicates the round-to-nearest operation, and $\text{clip}(\cdot, \cdot, \cdot)$ indicates the element-wise clipping operation, which ensures that $\boldsymbol{X}^{(int)}$ can be represented in the chosen bit-width [34]. The *dequantization* operation then produces a matrix $\boldsymbol{X}^{(q)}$ which is a quantized approximation of the input matrix $\boldsymbol{X}$: $\boldsymbol{X}^{(q)} = s\boldsymbol{X}^{(int)} \approx \boldsymbol{X}$. This allows the use of efficient integer matrix multiplication [34]. For a matrix $\boldsymbol{Y} \in \mathbb{R}^{n \times k}$,

$$\boldsymbol{X}\boldsymbol{Y} \approx \boldsymbol{X}^{(q)}\boldsymbol{Y}^{(q)} = s_x s_y \boldsymbol{X}^{(int)}\boldsymbol{Y}^{(int)}. \tag{1}$$

### 2.2 Floating point number system

A floating point number set $F \subset \mathbb{R}$ is a set whose elements are defined as follows:

$$f = (-1)^s 2^{p-b}\left(1 + \frac{d_1}{2} + \frac{d_2}{2^2} + \cdots \frac{d_m}{2^m}\right), \tag{2}$$

where $s \in \{0, 1\}$ is the sign bit, $d_i \in \{0, 1\}$ is the $m$-bit mantissa, $p \in \mathbb{Z}; 0 \leq p < 2^e$ is the $e$-bit exponent, and $b$ is an integer *exponent bias*, commonly defined to be $2^{e-1}$.

Floating point numbers can be seen as a uniform $m$-bit grid between two consecutive (integer) powers of two $2^a, 2^{a+1}$. The distance between grid points in the range $[2^a, 2^{a+1}]$ is $2^{a-m}$. Increasing the number of mantissa bits thus increases the number of grid points in each range $[2^a, 2^{a+1}]$. In the definition provided earlier in this section, the number of ranges $[2^a, 2^{a+1}]$ that can be represented by a floating point number system, is determined by the number of exponent bits $e$. Increasing the number of exponent bits thus increases the dynamic range (i.e. ratio between largest and smallest non-zero value) of values that can be represented. Any fixed bit-width floating point number system must make a trade-off between the dynamic range of representable values ($e$), and the precision

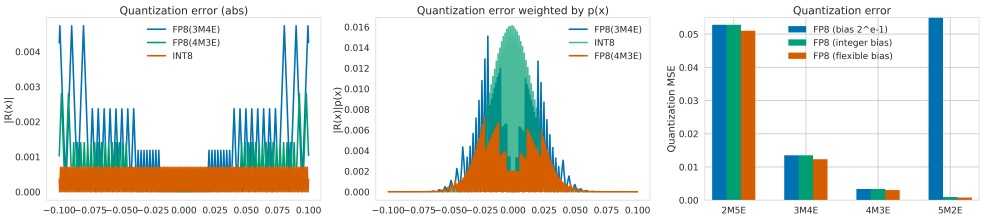

Figure 2: (left) Rounding error for quantization grids: FP8 with 4-bit exponent, FP8 with 3-bit exponent, INT8. (middle) The rounding error weighted by the probability density function from the first column. (right) Exponent bias ablation for a zero mean Gaussian distribution for FP8 format. We consider three versions of the format: fixed format with the bias $2^{e-1}$, integer bias, FP32 scale.

$(m)$. For example, IEEE-754 32-bit 'full-precision' floating point numbers (FP32) use 1 sign bit, 23 mantissa bits, and 8 exponent bits. The resulting effect is that, compared to integer formats, floating point formats have more precision close to zero, as the ranges $[2^a, 2^{a+1}]$ will be smaller for lower values of $a$, and less precision away from 0. This is visualized in the left plot in Fig. 1. Intuitively, they are a better match for peaked distributions like Gaussians that have more density around 0, and a better fit for distributions with large tails and outliers like the Student's t-distribution. The right plot in Fig. 1 shows the different distributions of values for various FP8 formats.

Note that this definition does not allow for a representation of 0. To allow 0 to be represented, the exponent value $p = 0$ is reserved to indicate *subnormal* numbers. In this case, the exponent value is implicitly set to 1, and $f = (-1)^s 2^{1-b} \left( 0 + \frac{d_1}{2} + \frac{d_2}{2^2} + \cdots \frac{d_m}{2^m} \right)$. Besides allowing the exact representation of 0, subnormal numbers also allow a graceful representation of values close to 0. See Fig. 1 for an intuition behind this. In Section 4 we show (de)quantization operations analogous to those of INT8.

## 2.3 Assumptions and extensions

In this work we make a number of assumptions and consider several extensions of the standard floating point format. We assume that the exponent bias $b$ is not restricted to the value $2^{e-1}$. Instead, we introduce a (per-tensor or per-channel) quantization scale $\gamma$, similar to the quantization scale used in integer quantization [34, 26]. Since there is no standard allocation of mantissa and exponent bits for 8 bit floating point formats, we consider various allocations of mantissa and exponent bits, as we assume that the choice of trade-off between precision and dynamic range will have more impact for lower floating point bit-widths. We use the notation xMyE to denote a format with x mantissa bits and y exponent bits. Specifically, we consider the formats 5M2E, 4M3E, 3M4E and 2M5E.

## 3 Expected quantization error

In this section we perform an analytical analysis of the FP8 and INT8 formats and show that formats with more exponent bits perform better when outliers are a factor. This theoretical result lays a foundation for the comparison between the formats that is predictive of when to use the FP8 format and how many exponent bits to use.

We investigate the error induced by quantizing values drawn from several insightful distributions, considering the expected MSE of these values as it has been shown to be indicative of the final loss in a neural network [33].

### 3.1 Expected quantization error

Given a quantization grid $\alpha = \{\alpha_1, \alpha_2, \ldots \alpha_k\}$, we can define the quantization operation $Q_\alpha(w)$ and the corresponding quantization error $R_\alpha(w)$:

$$Q_\alpha(w) = \alpha_i, \ i = \arg\min_i |w - \alpha_i|, \qquad R_\alpha(w) = Q_\alpha(w) - w. \qquad (3)$$

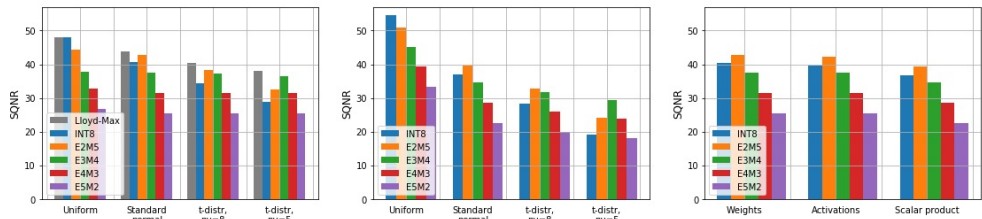

Figure 3: (left) Quantization expected SQNR for different distributions and different formats including Lloyd-Max non-uniform quantizer [30, 32]. The Student's-t distribution is clipped at $[-100, 100]$ to avoid an unbounded error. (middle) scalar product expected SQNR, (right) Expected SQNR for a Resnet18 layer.

Examples of the rounding error function for the 3M4E, 4M3E, and INT8 formats are shown in fig. 2 (left). We model neural network weights as a random variable $W \sim p_w(w)$. The expected value of the quantization MSE can be expressed as follows:

$$\mathbb{E}(R_\alpha(W))^2 = \int_{\alpha_{min}}^{\alpha_{max}} R_\alpha^2(w)p_w dw + \int_{-\infty}^{\alpha_{min}} (w - \alpha_{min})^2 p_w dw + \int_{\alpha_{max}}^{\infty} (\alpha_{max} - w)^2 p_w dw, \quad (4)$$

where $\alpha_{min} = \min_i \alpha_i$ and $\alpha_{max} = \max_i \alpha_i$ are the limits of the quantization range. The first term corresponds to the rounding error, while the remaining two terms correspond to the clipping error. We will use this formulation to analytically compute the quantization errors for different distributions. This is done by splitting the integration into sub-intervals corresponding to each point of the quantization grid. We present further details of this procedure the appendix A.1. An example of the rounding error function weighted by the probability density is given in fig. 2 (middle).

## 3.2 Scalar product quantization error

We can now model the expected output error for a quantized scalar product, the essential building-block for both fully-connected and convolutional layers. In order to model the activation values, we introduce a random variable $X \sim p_x(x)$ and the corresponding quantization grid $\beta = \{\beta_1, \beta_2, \ldots \beta_k\}$, $\beta_i \in \mathbb{R}$. While computing the scalar product of $X$ and $W$, we only consider the error incurred by quantization of the inputs, as the error incurred by quantization of the output $Y$ is identical to the input quantization error as described in the previous section.

We denote the output tensor $Y = WX$, and its quantized version as $Y_Q$. The quantization error $\Delta Y$ then can be expressed as

$$\Delta Y = Y_Q - Y = Q_\alpha(W)Q_\beta(X) - WX = WR_\beta(X) + XR_\alpha(W) + R_\beta(X)R_\alpha(W). \quad (5)$$

The sample mean value $\overline{\Delta Y^2}$ corresponds to the mean error of the scalar product between two vectors $\mathbf{W} \in \mathbb{R}^n$ and $\mathbf{X} \in \mathbb{R}^n$, where $W_i \sim p_w(w)$, $X_i \sim p_x(x)$ are weights and activations samples:

$$\overline{\Delta Y^2} = \frac{1}{n} \left[ \langle Q(\mathbf{W}), Q(\mathbf{X}) \rangle - \langle \mathbf{W}, \mathbf{X} \rangle \right]^2. \quad (6)$$

Computing $\mathbb{E}(\Delta Y^2)$ for different distributions and quantization grids allows us to analyze how they would work in practical scenarios. In many cases, the expected output MSE can be approximated by the following weighted sum of the rounded errors on $W$ and $X$:

$$\mathbb{E}(\Delta Y^2) \approx \mathbb{E}(R_\alpha(W))^2 \int_{-\infty}^{\infty} x^2 p_x(x)dx + \mathbb{E}(R_\beta(X))^2 \int_{-\infty}^{\infty} w^2 p_w(w)dw. \quad (7)$$

We see that the output error mainly depends on the quantization error of the inputs and the spread of the distributions of the inputs. We describe the full procedure for computing $\mathbb{E}(\Delta Y^2)$ and further analysis in appendix A.2.

**Discussion** We use these formulations to analytically compute the quantization errors for several distributions. We consider the uniform distribution, Gaussian distribution and the student-t distribution. The latter is essentially a more heavy-tailed Gaussian, allowing us to model what happens when outliers are added into the picture.

It is important to choose the bias term in the FP8 format properly. We see in fig. 2(right) that the standard fixed format of setting the bias to $2^{e-1}$ can fail. This happens when values are either clipped too much, or not enough grid-points are used as the representable range is too big. We also see that having an integer bias performs worse than having a floating-point bias. This is similar to what happens in INT8 quantization, where setting the scale parameter correctly can have a large impact on the network's performance [34]. As the bias shifts the dynamic range, this effect is less strong for the FP8 formats with a high amount of exponent bits that have a high dynamic range. For lower exponent-bits, one would preferably use a per-channel bias/scale parameter, similar to what is done in per-channel quantization [26]. Further justification for this can be found in section 5.2.

We also analyze the effect of different settings of the number of mantissa and exponent bits. The results of this are presented in fig. 3. We observe a clear pattern. For a uniform distribution, the INT8 format performs the best. For the Gaussian distribution, the format with 2 exponent bits performs best. Then, when increasing the relative outliers by decreasing the degrees of freedom in the Student's t-distribution, the best format tends towards having more exponent bits.

In neural networks, most weights and activations can be modeled relatively well by a Gaussian distribution. So based on this analysis, we would expect the 5M2E format to work the best for most well-behaved layers with little outliers. Specifically in the weights outliers are infrequent, as the weights are often explicitly, and otherwise implicitly regularized [45]. For example, we consider a layer of Resnet18 model pre-trained on ImageNet. We fit Gaussian distributions in the weights and activations sample, and compute the expected MSE analytically in fig. 3 (right). We see that the 5M2E format works the best in this case. A more detailed per-layer study is given in appendix B.1.

However, the stronger the outliers are in the distribution, the more the FP8 format with a higher number of bits will help with its quantized representation. We would thus predict that for networks with severe outliers, formats like M4E3 would perform better. We will show in section 5 that these predictions hold for real networks, and that networks that are known to have larger activation outliers such as transformers benefit the most from more exponent bits.

## 4 FP8 quantization simulation

In the previous section our analysis showed that FP8 quantization can theoretically yield better tensor reconstruction MSE than INT8 quantization, but careful selection of the division between exponent and mantissa bits, as well as the value of the exponent bias are crucial. We investigate whether these findings can be extended to FP8 quantization of neural networks. In order to do so, we need an efficient simulation of FP8 quantization. Methods commonly used for FP8 quantization are either too slow, e.g. nearest grid point search or bit-masking [22], or too complicated, e.g. custom CUDA kernels [40], for quick experimentation.

In this section we introduce a novel method of FP8 quantization simulation. This method can easily be implemented in common deep learning frameworks. An added benefit of this method is that it exposes the parameters of the FP8 quantizer (i.e., the number of mantissa/exponent bits and the exponent bias), thus allowing the parameters to be learned by back-propagation.

### 4.1 FP8 simulated quantization formulation

In our FP8 quantizer we exploit the fact that FP8 quantization can be seen as the union of $m$-bit uniform quantization grids between consecutive integer powers of two $[2^{p-b}, 2^{p+1-b}]$. This means we can simulate FP8 quantization of an input vector[2] $\boldsymbol{x}$ using the same method as for simulation of uniform quantization as described in Section 2.1, with the distinction that each element $x_i$ in $\boldsymbol{x}$ has its own associated scale $s_i$:

$$x_i^{(q)} = s_i \left\lfloor \frac{x_i}{s_i} \right\rceil, \tag{8}$$

---

[2]This method extends readily to matrices or higher-dimensional tensors.

where $x_i^{(q)}$ denotes $x_i$ quantized to floating point. The scale $s_i$ depends on the number of mantissa bits $m$ and the range $[2^{p-b}, 2^{p+1-b})$ in which $x_i$ falls. This is given by $\log_2 s_i = p_i = \lfloor \log_2 |x_i| \rfloor - m$.

To ensure that $x_i^{(q)}$ can be represented given $m$, $e$ and $b$, both $x_i^{(q)}$ and $s_i$ need to be clipped. Values of $x_i^{(q)}$ greater than maximum value $c$ or smaller than $-c$ are clipped $c$ and $-c$ respectively, where $c = (2 - 2^{-m})2^{2^e - b - 1}$ is the largest representable value for a given floating point format. Since $2^{1-b-m}$ is the smallest representable value, values of $p_i$ smaller than $1 - b - m$ are clipped to $1 - b - m$.

Note that this approach is identical to the quantization operation defined in Eq. 3, provided that the rounding mode in Eq. 8 matches the tie-breaking procedure in Eq. 3 for numbers equidistant to two numbers in $F$. See Fig. 4 for an intuition.

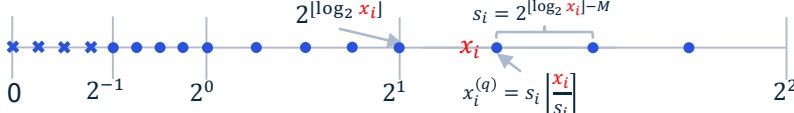

Figure 4: Graphic depiction of the FP8 quantizer.

In case the scaling factor $\gamma \neq 1$ this needs to be reflected in $p_i$. In order to accommodate the scaling factor, we first fold it into a reparameterized bias value $\widehat{b} = b - \log_2 \gamma$. We then compute $p_i$ as follows:

$$p_i = \begin{cases} \left\lfloor \log_2 |x_i| + \widehat{b} \right\rfloor - \widehat{b} - \lfloor m \rceil, & \text{if } \left\lfloor \log_2 |x_i| + \widehat{b} \right\rfloor > 1 \\ 1 - \widehat{b} - \lfloor m \rceil, & \text{otherwise.} \end{cases} \tag{9}$$

## 4.2 Quantization-aware training with FP8

To enable QAT using this quantizer, we need to make a few changes. First, to allow gradients to flow through each step of the quantizer, we use the straight-through estimator (STE) for gradients of non-differentiable rounding operations. Second, we find that learning the maximum clipping value $c$ instead of $\widehat{b}$ improves training stability. $\widehat{b}$ can be found from $c$ as follows: $\widehat{b} = 2^e - \log_2 c - \log_2(2 - 2^{-m}) - 1$. Lastly, we treat $\left\lfloor \log_2 |x_i| + \widehat{b} \right\rfloor$ as a constant that receives no gradient. This prevents the (sometimes extremely large) gradients of this operation w.r.t. $x_i$ to propagate backwards. The result of this is that $x$ receives the 'straight-through' gradient for the full quantization procedure, i.e. $\frac{\partial}{\partial x_i} F(x_i, m, c) = 1$, where $F(\cdot, \cdot, \cdot)$ denotes the FP8 quantizer.

## 4.3 Toy experiment: Learning minimal MSE on common distributions

To investigate whether our quantizer can indeed learn the maximum value $c$ and number of mantissa bits $m$, we run a toy experiment. We sample $10^5$ values from $\mathcal{N}(0, 1)$, and initialize an FP8 quantizer with 3M4E and a bias of 8, which corresponds to maximum value of $c = 240$. We then use SGD to learn the values of $c$ and $m$ that minimize MSE on the reconstruction loss: $\mathcal{L}(M, c) = \frac{1}{N} \sum_i (x_i - F(x_i, m, c))^2$. After 500 iterations, $c$ has converged to 4.35 and $m$ oscillates around 5.5. The oscillation behavior can be explained by the fact that MSE can be further minimized by increasing precision through higher $m$, however increasing $m$ to 6 yields a uniform quantizer, which results in higher MSE. Similar behavior was observed in uniform quantization by [35]. A line search shows that indeed $m = 5$ and $c = 4.37$ minimize the MSE on the target data, meaning our algorithm found the optimal values in this example. See Fig. 5 for details.

# 5 Experiments

In this section, we aim to empirically validate the analytical findings from section 3 on full neural networks. We use our analytical results to investigate various FP8 formats for weight and activation tensors in neural networks, and show that Finally, we perform quantization-aware training (QAT) on FP8 quantized models.

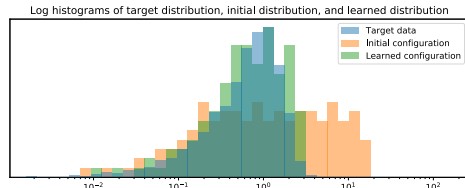 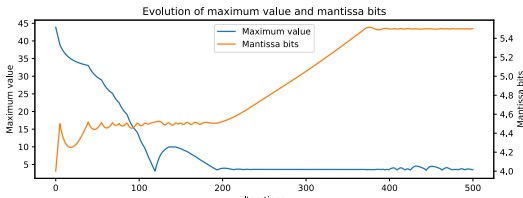

Figure 5: **Left**: Histograms of target data, and representable values of the initial format (3M4E) and the learned format (5M2E). **Right**: evolution of maximum value and number of mantissa bits. After roughly 350 iterations the number of mantissa bits starts oscillating around a value of 5.5.

## 5.1 Baselines and experimental setup

We run our experiments on a variety of different tasks, datasets, and models: We experiment on ResNet18 [19], MobileNetV2 [38], and ViT [14] for ImageNet classification [37]; BERT-base [12] for language understanding on the GLUE benchmark [43]; HRNet [39] for semantic segmentation on the Cityscapes dataset [10]; DeepLabV3 [7] for semantic segmentation on the Pascal VOC dataset [16]; and SalsaNext [11] for LIDAR point cloud segmentation on the SemanticKITTI dataset [2]. For each model, we report the 32-bit floating point (FP32) results on the same model instance (trained weights and code base) used in our other experiments.

As baselines we compare against models quantized using INT8 quantization. Following [35] we do not apply batch normalization folding, and re-estimate the batch normalization statistics (running mean and variance) before final validation, as this improved results for every model we considered. For each model, we consider several methods for range estimation separately for weights and activations, per-channel and per-tensor range estimation for weights. Results reported are those for the best combination of range estimation setting and per-channel or per-tensor range estimation. Our code is written in PyTorch and all our experiments are performed using NVIDIA Tesla V100 and A100 GPUs.

## 5.2 Post-training quantization results

We compare post-training quantization using INT8 and FP8 quantization. To allow an apples-to-apples comparison, we do not use any methods to improve post-training quantization that have been developed in the past years (e.g. those mentioned in [34]), other than the methods described in Section 5.1.

For our FP8 results, we report the best fully fixed format, i.e. the same combination of $m, e$ and $b$ used for the full network, the best flexible bias format, i.e. $m$ and $e$ fixed for the full network, but $\widehat{b}$ applied per channel or tensor, whichever yields best results, and a fully flexible format where $m$ and $e$ are set for each tensor, and $\widehat{b}$ is set for each channel, to minimize MSE. The full procedure for this approach is detailed in Section E. For an example setting, see Figures 11 and 12 for an example of FP8 formats that minimize MSE for each tensor in ResNet18 and BERT, respectively.

**Discussion**  PTQ results can be found in Table 1. In this table we can see that networks with large activation outliers (ViT, Bert, SalsaNext and HRNet) show better results for formats with more exponent bits, and thus a larger dynamic range. Furthermore, we see that, as predicted in Section A.1, formats with more mantissa bits are better able to represent weight and activation tensors in convolutional neural networks. However, since increasing the number of mantissa bits implies reducing dynamic range, finding the right value for the bias for each channel is important. The full set of results for our PTQ experiments is detailed in Table 3 in Appendix Section I.

Lastly, we see that fully flexible formats only sometimes outperform fixed $m/e$ formats, with slim improvements when they do. This is surprising, as a more flexible format should be able to better fit a network's tensors. We attribute this discrepancy to our local greedy method for assigning $m, e$ and $\widehat{b}$, which may not find settings that are optimal globally.

| Model | FP32 | INT8 | Best fixed | | Best flex bias | | Best flexible |
|---|---|---|---|---|---|---|---|
| | | | Format | Score | Format | Score | Score |
| ResNet18 | 69.72 | 69.55 | 3M4E9B | 68.52 | 5M2E | 69.66 | 69.64 |
| MobileNetV2 | 71.70 | 70.94 | 4M3E3B | 68.49 | 5M2E | 71.06 | 71.28 |
| ViT | 77.75 | 76.39 | 3M4E8B | 77.37 | 4M3E | 77.71 | 77.62 |
| BERT-base | 83.06 | 71.03 | 3M4E9B | 83.08 | 3M4E | 82.80 | 83.02 |
| SalsaNext | 55.80 | 54.22 | 4M3E3B | 55.79 | 4M3E | 55.80 | 55.02 |
| DeepLabV3 | 72.91 | 71.24 | 3M4E8B | 9.42 | 5M2E | 72.58 | 72.72 |
| HRNet | 81.05 | 80.93 | 3M4E8B | 80.71 | 4M3E | 81.04 | 81.06 |

Table 1: PTQ results with for fixed formats, flexible bias formats, and fully flexible formats, compared to original model (FP32) and INT8 results.

| Model | FP32 | INT8 | Best fixed accuracy | Best flex bias accuracy | Best flexible accuracy |
|---|---|---|---|---|---|
| ResNet18 PTQ | | 69.55 | 68.44 | 69.66 | 69.64 |
| ResNet18 QAT | 69.72 | 70.43 | 69.16 | 69.82 | 70.28 |
| + $c$ learning | | | 69.30 | 69.97 | 70.15 |
| + $m$ learning | | | 70.10 | 70.25 | 70.24 |
| MobileNetV2 PTQ | | 70.94 | 68.49 | 71.06 | 71.28 |
| MobileNetV2 QAT | 71.70 | 71.82 | 71.03 | 71.54 | 71.60 |
| + $c$ learning | | | 70.93 | 71.67 | 71.66 |
| + $m$ learning | | | 71.34 | 71.69 | 71.74 |
| BERT PTQ | | 71.03 | 83.08 | 82.80 | 83.02 |
| BERT QAT | 83.06 | 83.26 | 83.56 | 83.70 | 83.17 |
| + $c$ learning | | | 83.89 | 83.86 | 83.72 |
| + $m$ learning | | | 83.85 | 83.84 | 83.70 |

Table 2: QAT results with for fixed formats, flexible bias formats, and fully flexible formats, compared to original model (FP32) and INT8 QAT results.

### 5.3 Quantization-aware training

We investigate whether we can improve on the FP8 PTQ results by performing training with FP8 quantization in mind (quantization-aware training, QAT). We compare against INT8 QAT, which has previously been shown to result in INT8 models with accuracy near the original FP32 models.

We consider three different initializations based on our PTQ results from Table 1: best fixed format, best flexible bias format and best fully flexible format. For each of these initializations we perform QAT with the format fixed at initialization, with learnable maximum value $c$ (cf. range learning in INT8 QAT), and with learning both $c$ and the number of mantissa bits $m$. We run experiments with various learning rates for model and quantization parameters, as well as per-tensor and per-channel quantization, and report results for the best learning setup. We train our models for 20 epochs and use Adam for the model parameters and SGD for the quantization parameters. Results of these experiments can be found in Table 2. Full experimental details and results can be found in Appendix G.

**Discussion** From these results, we see that QAT can improve performance of FP8 quantized models. We also see that, for ResNet18 and MobileNetV2 learning both maximum value $c$ and number of exponent bits $m$ improves results compared to just learning weights, for fully fixed and flexible bias formats. See also Figure 6 for an example of how initialized $c$ and $m$ differ from learned $c$ and $m$ after 20 epochs of training. However, this difference disappears for fully flexible PTQ initialization, and learning $c$ and $m$ slightly harms performance for BERT, although the difference is too small to be significant.

Generally, we see that QAT reduces the accuracy gap between different formats, as the weights of the network can adapt to the distributions the quantizers represent.

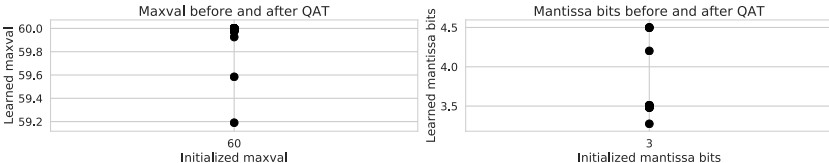

Figure 6: Values of maximum value $c$ and number of mantissa bits $m$ before and after training. In this experiment, maximum value was initialized to $c = 60$, and number of mantissa bits was initialized to $m = 3$.

# 6   Related work

**Integer quantization**   Integer quantization, sometimes also called fixed-point quantization, has been an active field of research since neural networks became more popular and the need for efficient inference became important. On a high level there are two lines of work, post-training quantization (PTQ) [29, 36, 13, 1, 9, 6, 33, 28] and quantization-aware training (QAT) [18, 23, 46, 8, 31, 15, 24, 5, 42, 35]. For more details we refer the interested read to the surveys of [27, 34, 17]. Our approach to learning the FP8 configuration is inspired by [15] which assume the straight through estimator (STE) [4] on the rounding operation in order to derive a gradient for the scaling factor of the fixed point format. [42] extends this to jointly learn the bit width and scaling factor, which we explore for learning the trade-off between the number mantissa and exponent bits.

**Floating point formats**   The accuracy and stability of floating point formats and algorithms are well studied in computer science and electrical engineering, we refer the interested read to [20]. However, these studies usually focus on high bit with floating point formats (e.g. FP32 or FP64) and do not consider the impact on neural networks which are significantly more robust to noise.

Early work by [44] showed that convolutional networks can be trained using FP16 and FP8 using the 2M5E format. The work of [40] introduces a hybrid FP8 format and first discovered that 3M4E has better inference performance than 2M5E. While they use a different FP8 format for the backwards path, they keep the bias and mantissa and exponent division fixed for all layers in the forward path. [22] introduces a fully flexible FP8 format similar to the one we consider. They perform a layer-wise optimization to find the best configuration (mantissa, exponent, sign bit, bias). Their work does not go in-depth in the analysis of why some formats are better, and it has a fairly limited results section with easy to quantize models like ResNet18 and VGG [27]. [41] uses a custom FP4 format for the backwards path and INT4 for the forward path to enable full 4 bit training with a small drop in performance.

To the best of our knowledge we are the first work with an extensive study of the different FP8 formats based on both analytical insights and empirical results across several tasks and data modalities. As opposed to our new FP8 simulation method, other works rely on dedicated FP8 simulations that do not allow for efficient gradient based learning of the bias or mantissa/exponent bit configuration.

# 7   Impact and Limitations

**Impact**   FP8 is becoming widespread[3] as an alternative to INT8, with multiple chips from vendors like Nvidia, Intel, Graphcore, AMD and IBM moving to support 3M4E and/or 2M5E FP8 formats, often with fixed bias values. In this paper we show that these formats are not optimal and that many networks benefit from FP8 formats with more mantissa bits and bias values that can be flexibly set per channel. We hope that the results in this paper will help guide FP8 hardware design decisions in the broader community.

**Limitations**   In this paper, we restrict ourselves to studying the impact of various FP8 formats on model accuracy and ignore hardware implementation-specific impact on power consumption and latency. Assessing the difference in hardware impact between INT8 and FP8 for all networks is not trivial. From a data transfer bandwidth perspective, the two 8-bit formats incur similar overhead, while

---

[3]http://bit.ly/3Sd8wey

for compute limited models, the relative overheads depend on the exact implementation. Generally, FP8 units use more power for additions and multiplications [21], however, surrounding logic (e.g. accumulator design, NaN/overflow checks etc) might amortize this difference, and in multi-purpose designs which support not only FP8, but e.g., also FP16/32, INT8/16 the difference in the overheads might disappear. Therefore we limited this work to comparing the different formats on a bit width level. Hardware design teams use our accuracy analysis to make trade-offs on the hardware side for their specific use-case and design.

## 8  Conclusion

In our analysis of the FP8 format and its benefits for quantized neural network inference we have touched on many points. We have shown that analytically the FP8 format can improve on the INT8 format for Gaussian distributions that are common in neural networks, and that higher exponent bits work well when outliers occur. This paper introduced a new way to simulate FP8 quantization in FP32 hardware that speeds up FP8 quantization simulation and makes it possible to learn the bias and mantissa-exponent bit-width trade-off. We validated the FP8 format for many networks in a post-training quantization setting, showing that generally for neural networks the 5M2E and 4M3E FP8 format works the best, and that for networks with more outliers like transformers increasing the number of exponent bits works best. We have also shown that when doing quantization-aware training, many of these benefits of the format disappear, as the network learns to perform well for the INT8 quantization grid as well.

## Acknowledgments

We would like to thank Christos Louizos and Yin Huang for helpful discussions and valuable feedback.

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
