# A   Analytical computation of the expected quantization error

## A.1   Input quantization error.

Equation 4 (the quantization error) can be split into two terms corresponding to the rounding error $E_{rw}$ and the clipping error $E_{cw}$:

$$\mathbb{E}(W - Q_\alpha(W))^2 = \mathbb{E}(R_\alpha(W))^2 = E_{rw} + E_{cw}, \tag{10}$$

$$E_{rw} = \int_{\alpha_{min}}^{\alpha_{max}} R_\alpha^2(w)p_w(w)dw, \tag{11}$$

$$E_{cw} = \int_{-\infty}^{\alpha_{min}} (w - \alpha_{min})^2 p_w(w)dw + \int_{\alpha_{max}}^{\infty} (\alpha_{max} - w)^2 p_w(w)dw. \tag{12}$$

As the weight and activation tensor values are bound, we assume that the distribution $p_w(w)$ is clipped within the interval $(w_{min}, w_{max})$. Thus the clipping error can be written as:

$$E_{cw} = \mathbb{1}_{w_{min} < \alpha_{min}} \int_{w_{min}}^{\alpha_{min}} (w - \alpha_{min})^2 p_w(w)dw + \mathbb{1}_{\alpha_{max} < w_{max}} \int_{\alpha_{max}}^{w_{max}} (\alpha_{max} - w)^2 p_w(w)dw, \tag{13}$$

where $\mathbb{1}_{\alpha_{max} < w_{max}}$ is the indicator function. The calculation or the rounding error $E_{rw}$ can be split into two sub-intervals for each interval $(\alpha_i, \alpha_{i+1})$ where the first sub-interval corresponds to rounding up and the second sub-interval corresponds to rounding down:

$$E_{rw} = \sum_{i=1}^{|\alpha|} \int_{\alpha_i}^{\alpha_{i+1}} R_\alpha^2(w)dw = \sum_{i=1}^{|\alpha|} \int_{\alpha_i}^{(\alpha_i + \alpha_{i+1})/2} (w - \alpha_i)^2 p_w(w)dw + $$
$$\sum_{i=1}^{|\alpha|} \int_{(\alpha_i + \alpha_{i+1})/2}^{\alpha_{i+1}} (\alpha_{i+1} - w)^2 p_w(w)dw. \tag{14}$$

In order to simplify the computation, we introduce the following function:

$$I_w(a, b, w_0) := \int_a^b (w - w_0)^2 p_w(w)dw. \tag{15}$$

Thus we can redefine 14 as:

$$E_{rw} = \sum_{i=1}^{|\alpha|} \left[ I(\alpha_i, (\alpha_i + \alpha_{i+1})/2, \alpha_i) + I((\alpha_i + \alpha_{i+1})/2, \alpha_{i+1}, \alpha_{i+1}) \right]. \tag{16}$$

The sum of the rounding errors for each interval between two representable grid-points. We note that the clipping error $E_{cw}$ can also be expressed using $I_w(a, b, w_0)$:

$$E_{cw} = I_w(w_{min}, \alpha_{min}, \alpha_{min}) + I_w(\alpha_{max}, w_{max}, \alpha_{max}). \tag{17}$$

The analytical expressions for $I(w_{min}, \alpha_{min}, \alpha_{min})$ for different distributions are given in Appendix A.3. Thus, given the explicit definition of the quantization grid and the probability density function, we can analytically compute the rounding error for different distributions, for example the Gaussian, Uniform, or Student's t-distribution.

## A.2 Scalar product quantization error.

In this section, we describe a practical way for analytical computation of the expected MSE of the scalar product with quantized inputs $\mathbb{E}(\Delta Y^2)$. Practically, the covariance matrix for $W$ and $X$ is diagonally dominant therefore we can assume their independence. Thus the expected value $\mathbb{E}(\Delta Y^2)$ can further be expressed as:

$$\mathbb{E}(\Delta Y^2) = \int_{-\infty}^{\infty} \int_{-\infty}^{\infty} \left[ xR_\alpha(w) + wR_\beta(x) + R_\alpha(w)R_\beta(x) \right]^2 p_x(x)p_w(w)dwdx, \qquad (18)$$

$$\mathbb{E}(\Delta Y^2) = \int_{-\infty}^{\infty} x^2 p_x(x)dx \int_{-\infty}^{\infty} R_\alpha^2(w)p_w(w)dw + \int_{-\infty}^{\infty} w^2 p_w(w)dw \int_{-\infty}^{\infty} R_\beta^2(x)p_x(x)dx +$$

$$\int_{-\infty}^{\infty} R_\beta^2(x)p_x(x)dx \int_{-\infty}^{\infty} R_\alpha^2(w)p_w(w)dw + 2\int_{-\infty}^{\infty} wR_\alpha(w)p_w(w)dw \int_{-\infty}^{\infty} xR_\beta(x)p_x(x)dx + \quad (19)$$

$$2\int_{-\infty}^{\infty} R_\beta^2(x)p_x(x)dx \int_{-\infty}^{\infty} wR_\alpha(w)p_w(w)dw + 2\int_{-\infty}^{\infty} R_\alpha^2(w)p_w(w)dw \int_{-\infty}^{\infty} xR_\beta(x)p_x(x)dx.$$

We note that the first term is nothing but the rounding error on $W$ (see equation (4)) weighted by a non-central second moment on $X$ which does not depend on the quantization grid. The structure of the second term is very similar while $W$ and $X$ are interchanged. As the first two terms are the only integrals of non-negative functions, in practice they become dominant and mostly determine the MSE magnitude.

In order to compute the scalar product error analytically, we rewrite equation (19) in the following form:

$$\mathbb{E}(\Delta Y^2) = M_x E_{rw} + M_w E_{rx} + E_{rw}E_{rx} + 2E_{sw}E_{sx} + 2E_{rw}E_{sx} + 2E_{rx}E_{sw}, \qquad (20)$$

Where every term is introduced below. $M_w$ and $M_w$ are the second non-central moments for $W$ and $X$, respectively. These terms can be computed using functions $I_w(a, b, w_0)$ and $I_x(a, b, x_0) := \int_a^b (x - x_0)^2 p_x(x)dx$:

$$M_w = \int_{w_{min}}^{w_{max}} w^2 p_w(w)dw = I_w(w_{min}, w_{max}, 0),$$

$$M_x = \int_{x_{min}}^{x_{max}} x^2 p_x(x)dx = I_x(x_{min}, x_{max}, 0); \qquad (21)$$

$E_{rw}$ and $E_{rx}$ are rounding errors on $W$ and $X$ similar to the expression in equation (4):

$$E_{rw} = \int_{w_{min}}^{w_{max}} R_\alpha^2(w)p_x(w)dw,$$

$$E_{rx} = \int_{x_{min}}^{x_{max}} R_\beta^2(x)p_x(x)dx; \qquad (22)$$

finally, $E_{sw}$ and $E_{sx}$ are the following integrals.

$$E_{sw} = \int_{w_{min}}^{w_{max}} w R_\alpha(w) p_w(w) dw,$$

$$E_{sx} = \int_{x_{min}}^{x_{max}} x R_\beta(x) p_x(x) dx. \tag{23}$$

Similar to rounding error calculation in equation 14, the computation of $E_{sw}$ and $E_{sx}$ can be split into sub-intervals. For $E_{sw}$:

$$E_{sw} = \sum_{i=1}^{|\alpha|} \left[ \int_{\alpha_i}^{(\alpha_i+\alpha_{i+1})/2} w(w - \alpha_i) p_w(w) dw + \int_{(\alpha_i+\alpha_{i+1})/2}^{\alpha_{i+1}} w(\alpha_{i+1} - w) p_w(w) dw \right]. \tag{24}$$

We define $J_w(a, b, w_0)$ as follows:

$$J(a, b, w_0) := \int_a^b w(w - w_0) p(w) dw. \tag{25}$$

Thus we can express the term $E_{sw}$ in equation 24 as:

$$E_{sw} = \sum_{i=1}^{|\alpha|} \left[ J(\alpha_i, (\alpha_i + \alpha_{i+1})/2, \alpha_i) - J((\alpha_i + \alpha_{i+1})/2, \alpha_{i+1}, \alpha_{i+1}) \right]. \tag{26}$$

The term $E_{sx}$ can be computed in a similar way. The formulas for $J_w(a, b, w_0)$ for different distributions are given in Appendix A.3.

### A.3   Formulas for integrals $I_x(a, b, x_0)$ and $I_w(a, b, x_0)$.

In this section we give formulas for the functions $I_x(a, b, x_0)$ and $J_x(a, b, x_0)$ for different distributions which are necessary for the analytical computation of the rounding error and the scalar product error. The formulas for $I_w(a, b, w_0)$ and $J_w(a, b, w_0)$ are similar while $p_w(w)$ is used as the probability density function. The formulas were obtained using symbolical computations.

**Gaussian distribution.**

$$p(x) = \frac{1}{Z} \exp \left[ -\frac{1}{2} \left( \frac{x - \mu}{\sigma} \right)^2 \right], \tag{27}$$

where

$$Z = \frac{1}{\sigma \sqrt{2\pi}}, \tag{28}$$

$$
\begin{aligned}
I_x(a, b, x_0) = & -\exp \left( \frac{-a^2/2 + a\mu - \mu^2/2}{\sigma^2} \right) \sigma^2(-a - \mu + 2x_0)/Z + \\
& \sqrt{\frac{\pi}{2}} \sigma(-\mu^2 - \sigma^2 + 2\mu x_0 - x_0^2) \operatorname{erf}\left[ \frac{-a + \mu}{\sigma\sqrt{2}} \right] /Z + \\
& \exp \left[ \frac{-b^2/2 + b\mu - \mu^2/2}{\sigma^2} \right] \sigma^2(-b - \mu + 2x_0)/Z + \\
& \sigma\sqrt{\frac{\pi}{2}}(-\mu^2 - \sigma^2 + 2\mu x_0 - x_0^2) \operatorname{erf}\left[ \frac{-b + \mu}{\sigma\sqrt{2}} \right] /Z,
\end{aligned}
$$

$$J_x(a, b, x_0) = x_0 \sigma \exp\left[-\frac{\mu^2}{2\sigma^2}\right] \exp\left[\frac{(-a/2 + \mu)a}{\sigma^2}\right] / Z -$$

$$x_0 \sigma \exp\left[-\frac{\mu^2}{2\sigma^2}\right] \left(\exp\left[\frac{(-b/2 + \mu)b}{\sigma^2}\right] \sigma - \sqrt{\frac{\pi}{2}} \mu \operatorname{erf}\left[\frac{a - \mu}{\sqrt{2}\sigma}\right] + \sqrt{\frac{\pi}{2}} \mu \operatorname{erf}\left[\frac{b - \mu}{\sqrt{2}\sigma}\right]\right) / Z -$$

$$\frac{\sigma}{Z} \left(\exp\left[\frac{-a^2/2 + a\mu - \mu^2/2}{\sigma^2}\right] \sigma(a + \mu) + \sqrt{\frac{\pi}{2}}(\mu^2 - \sigma^2) \operatorname{erf}\left[\frac{\mu - a}{\sqrt{2}\sigma}\right]\right) +$$

$$\frac{\sigma}{Z} \left(\exp\left[\frac{-b^2/2 + b\mu - \mu^2/2}{\sigma^2}\right] \sigma(b + \mu) + \sqrt{\frac{\pi}{2}}(\mu^2 - \sigma^2) \operatorname{erf}\left[\frac{\mu - b}{\sqrt{2}\sigma}\right]\right).$$

$$(29)$$

**Uniform distribution.**

$$p(x) = \begin{cases} p_0, & \text{if } a \leq x \leq b \\ 0, & \text{otherwise,} \end{cases} \tag{30}$$

$$I_x(a, b, x_0) = p_0 \left(-\frac{a^3 + b^3}{3} + (a^2 - b^2)x_0 + (b - a)x_0^2\right), \tag{31}$$

$$J_x(a, b, x_0) = \frac{a^2 - b^2}{2} p_0 + (b - a)p_0 x_0. \tag{32}$$

**Student's t-distribution.**

$$p(x) = \frac{1}{Z} \left(1 + \frac{t^2}{\nu}\right)^{-(\nu+1)/2}, \tag{33}$$

$$Z = \sqrt{\nu} \mathcal{B}\left(\frac{1}{2}, \frac{\nu}{2}\right), \tag{34}$$

$$I_x(a, b, x_0) = \frac{2\nu x_0 (-1 + (\frac{a^2 + \nu}{\nu}))^{\frac{1-\nu}{2}}}{(1 - \nu)Z} - \frac{2\nu x_0 (-1 + (\frac{b^2 + \nu}{\nu}))^{\frac{1-\nu}{2}}}{(1 - \nu)Z} +$$

$$-au^2 \text{hyp2f1}(1/2, (1 + \nu)/2, 3/2, -a^2/\nu)/Z +$$

$$bu^2 \text{hyp2f1}(1/2, (1 + \nu)/2, 3/2, -b^2/\nu)/Z -$$

$$a^3 \text{hyp2f1}(3/2, (1 + \nu)/2, 5/2, -a^2/\nu)/(3Z) +$$

$$b^3 \text{hyp2f1}(3/2, (1 + \nu)/2, 5/2, -b^2/\nu)/(3Z), \tag{35}$$

where $\text{hyp2f1}(a, b, c, d)$ is Gaussian hypergeometric function.

$$J_x(a, b, x_0) = \frac{\nu^{(1+\nu)/2} x_0}{1 - \nu} \left[-(a^2 + \nu)^{(1-\nu)/2} + (b^2 + \nu)^{(1-\nu)/2}\right] / Z + I_x(a, b, 0). \tag{36}$$

# B Ablations

## B.1 Quantization error ablation

In this section we analyze different combinations of floating point formats for weights and activations based on the analytical computation of the scalar product error. We consider two Gaussian distributions fit into weights and activations distributions of a layer of pre-trained Resnet18 model. The results are shown in fig. 7. We observe that the optimal format for both weights and activations is

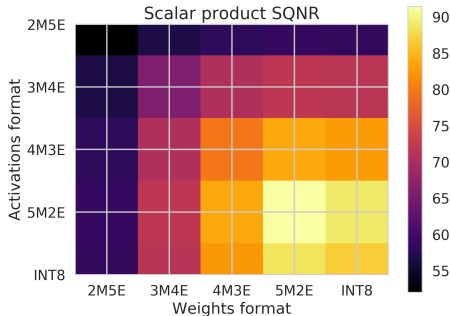

Figure 7: Study of a layer of pre-trained Resnet18 model. We fit two Gaussian distributions into weights and activations sample. Weights $W \sim \mathcal{N}(-1.0 \times 10^{-3}, 1.7 \times 10^{-2})$, clipped at $[-0.35, 0.35]$, activations $X \sim \mathcal{N}(0.06, 0.11)$ clipped at $[0, 3.63]$. The optimal format is M5E2 for both weights and activations.

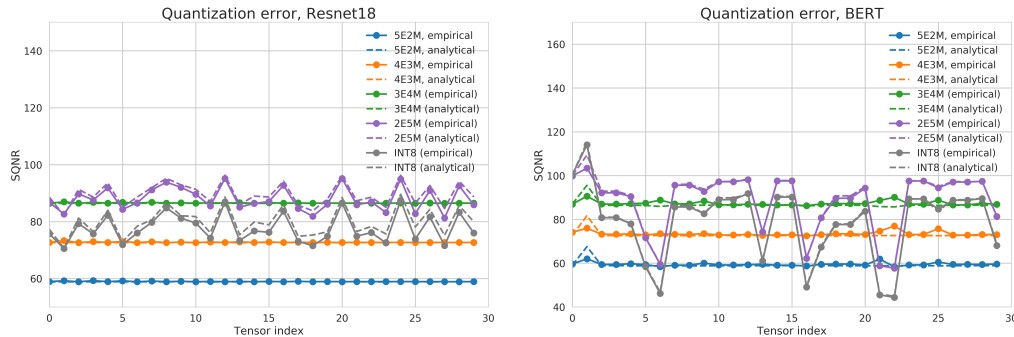

Figure 8: Comparison of the expected quantization error computed analytically to empirical quantization error. A random subset of 30 weights/activations tensor is chosen for Resnet18 and Bert.

.

5M2E. In order to facilitate visual comparison of MSE values of different magnitude we plot SQNR which is defined as follows:

$$10 \log_{10} \left( \frac{\mathbb{E}\left[\boldsymbol{W}^2\right] \mathbb{E}\left[\boldsymbol{X}^2\right]}{\mathbb{E}\left[\left(\boldsymbol{W}\boldsymbol{X} - Q_\alpha(\boldsymbol{W})Q_\beta(\boldsymbol{X})\right)^2\right]} \right). \tag{37}$$

### B.2 Comparison of the analytical and empirical rounding error.

In this section we compare the expected rounding error computed analytically to the empirical rounding error. The results are given on fig. 8 Note that, for visual purposes, we plot the signal-to-quantization-noise ratio (SQNR) instead of MSE in these plots. SQNR is log-proportional to MSE; i.e. a value that minimizes MSE will maximize SQNR. SQNR for an input tensor $\boldsymbol{X}$ is defined as:

$$10 \log_{10} \left( \frac{\mathbb{E}\left[\boldsymbol{X}^2\right]}{\mathbb{E}\left[\left(\boldsymbol{X} - Q_\alpha(\boldsymbol{X})\right)^2\right]} \right). \tag{38}$$

## C Examples of tensors in Bert

In this section we give examples of strong outliers in the tensors in Bert (fig. 9).

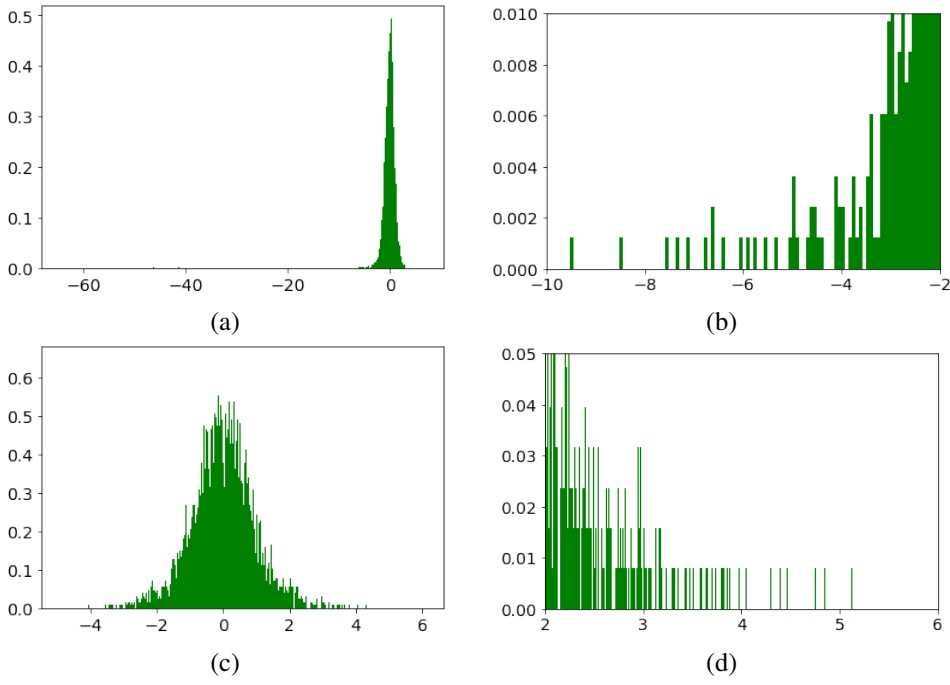

Figure 9: Examples of tensors in the forward pass of Bert. The plots (b) and (d) are scaled versions of the plots (a) and (b) respectively. The tails in the distributions contain significant amount of outliers.

# D   Importance of the outliers

In this ablation we demonstrate influence of the outliers on the choice of the optimal floating point format. The experiment is based on the analytical computation of the rounding error. We consider a Student's-t distribution with $\nu = 2$ with increasing quantization range. Min-max quantizer range estimator is used, the distribution is clipped at the quantization range. The results are given on fig. 10. While the quantization range is being increased, the optimal exponent bit-width values grows starting from zero (INT8 format) to 5-bit exponent.

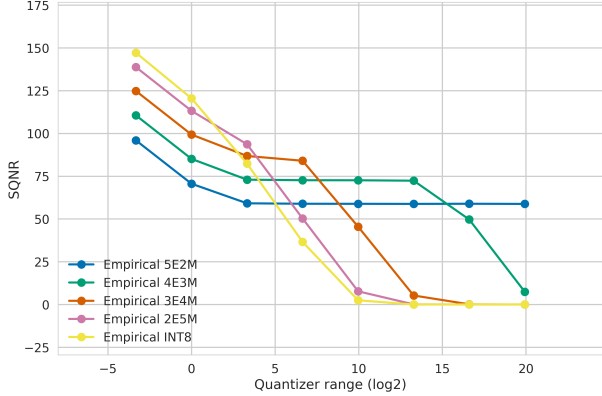

Figure 10: SQNR for Student's-t distribution with $\nu = 2$ for different quantizer range. Optimal exponent bit-width values increases while the quantization range is being expanded.

# E    MSE-based mantissa bits and bias

In our quantization setup, each tensor has an individual quantizer. The quantizers can use per-tensor or per-channel quantization. In case per-channel quantization is used, each channel has its own clipping value $c$ (and thus its own value for $\widehat{b}$), while the number of mantissa bits is set for a full tensor, and is thus shared across all channels.

During quantizer initialization we use the input weight tensor or one batch of activations, both referred to as $\boldsymbol{X}$, and find values for $m$ and $c$ that minimize reconstruction MSE. In order to do so, we perform a grid search over values of $m$ and values of $c$. For $m$ we consider 1, 2, 3, 4, 5, and 6 bits. For $c$ we find the absolute maximum value $\sigma$ of $\boldsymbol{X}$ (or, in case of per-channel quantization, each channel in $\boldsymbol{X}$), and run the grid search over 111 evenly spaced values between $0.1\sigma$ and $1.2\sigma$. We then record the values of $m$ and $c$ that minimize MSE and initialize the quantizer with these values.

In case per-channel quantzation is used, we run this procedure for each channel individually. On each channel, for each value of $m$, we store the value of $c$ that minimized quantization on that channel. We then choose a per-tensor value of $m$ by majority vote, i.e. the value of $m$ that occurred most often over all channels. In case of a tie we choose the value of $m$ that has lowest cumulative MSE. Lastly, we set $c$ to the value that minimized MSE for the per-tensor value of $m$. We also experimented with choosing $m$ based on lowest cumulative MSE directly, but found negligible difference in resulting accuracy. We decided to use majority vote to ensure channels with relatively large magnitude to dominate the choice of $m$. Figures 11 and 12 show a per-layer analysis of the bitwidth choices that minimize MSE.

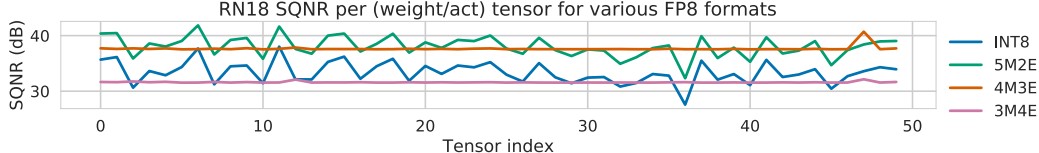

Figure 11: SQNR per layer for ResNet18.

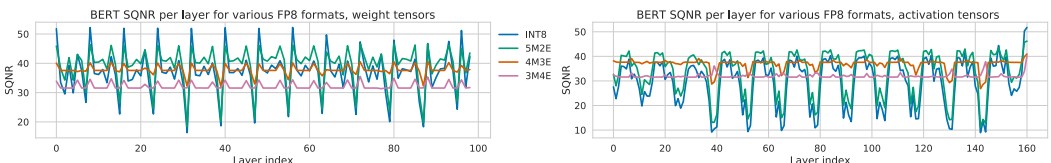

Figure 12: SQNR per layer for BERT (split up into weights and activations).

# F    Correlation between MSE in the output activations and the model accuracy

In this section we provide an example of a correlation between MSE in output activations of a layer and the full model accuracy. We take a pre-trained Resnet18 model, and consider one if its layers. We inject Gaussian noise of increasing amplitude in its weight values, and measure MSE in the output activations of the layer, and the final top-1 classification accuracy. The results are shown in figure 13. The MSE value and the final accuracy exhibit strong correlation, i.e. the normalized correlation coefficient value for this experiment is 0.98.

# G    Full experimental details

For our QAT experiments, we initialized our models using the settings that gave the best results for the fixed format, the flexible bias format, and the fully flexible format. See Table 1 and the tables in I for details. We then trained ResNet18 for 20 epochs, and MobileNetV2 for 10. We used the Adam [25] to optimize the weights. We considered starting learning rates of $10^{-5}$ and $10^{-6}$ as these gave

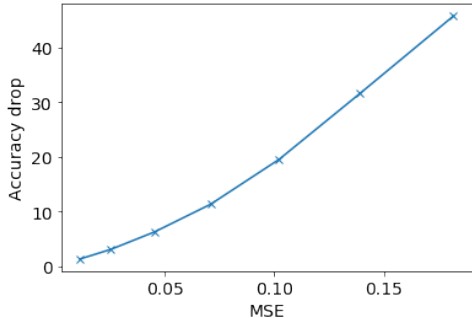

Figure 13: An ablation performed on one layer of Resnet18 pre-trained on ImageNet. Gaussian noise of increasing amplitude is added to the weights of the layer. The MSE in the output activations has strong correlation in the drop in top-1 accuracy of the model.

best results in a pilot experiments. Learning rates were decayed to a factor of $10^{-2}$ of the starting learning using cosine decay.

In experiments where FP8 parameters ($c$ and $m$) were learned as well, we used SGD without momentum. We considered learning rates $10^{-2}$, $10^{-3}$, $10^{-4}$, and $10^{-5}$. No weight decay was applied to any of the models.

Baseline INT8 QAT results followed the procedure as described in [34].

## H   Gradients of the FP8 quantizer

As stated previously, the FP8 quantizer gives the 'straight-through' gradient [3] w.r.t. the values to be quantized:

$$\frac{\partial}{\partial x_i} F(x_i, m, c) = \begin{cases} 1, & \text{if } -c \leq x_i \leq c \\ 0, & \text{otherwise.} \end{cases} \tag{39}$$

The gradient w.r.t. $c$ is as follows:

$$\frac{\partial}{\partial c} F(x_i, m, c) = \begin{cases} \frac{2^{p_i}}{c} \left( \left\lfloor \frac{x_c}{s} \right\rceil - \frac{x_c}{s} \right), & \text{if } \left\lfloor \log_2 |x_i| + \widehat{b} \right\rfloor > 1 \text{ and } -c \leq x_i \leq c, \\ \frac{1}{c \ln(2)}, & \text{if } \left\lfloor \log_2 |x_i| + \widehat{b} \right\rfloor \leq 1 \text{ and } -c \leq x_i \leq c, \\ -1, & \text{if } x_i < -c, \\ 1, & \text{if } x_i > c, \end{cases} \tag{40}$$

Lastly, the gradient w.r.t. $m$ is as follows:

$$\frac{\partial}{\partial m} F(x_i, m, c) = 2^{p_i} \ln(2) \left( \left\lfloor \frac{x_c}{s} \right\rceil - \frac{x_c}{s} \right) \left( 2^{7 - \lfloor m \rceil} + \frac{2^{-m}}{(2 - 2^{-m})} \right) \tag{41}$$

## I   PTQ results

The best results for INT8, FP8 with fixed bias (per Mantissa/Exponent division), FP8 with flexible bias, and fully flexible FP8, on all models considered in Section 5.2 are shown in Table 3.

Per-task BERT results are included in Table 4.

## J   DeepLabV3 weight distribution

DeepLabV3 shows a larger degradation in the 'fixed format' PTQ setting than other models considered. Figure 14 shows the distribution of the values in each weight tensor in DeepLabV3. We

| | FP32 | INT8 | Fixed bias | | | Flex bias | | | | Flex |
|---|---|---|---|---|---|---|---|---|---|---|
| | | | 5M2E | 4M3E | 3M4E | 5M2E | 4M3E | 3M4E | 2M5E | |
| ResNet18 | 69.72 | 69.64 | 27.25 | 68.35 | **68.44** | **69.66** | 69.45 | 68.57 | 64.92 | 69.64 |
| MobileNetV2 | 71.70 | 70.94 | 20.35 | **68.49** | 66.50 | **71.06** | 70.62 | 66.05 | 49.51 | **71.28** |
| ViT | 77.75 | 76.41 | 51.05 | 76.97 | **77.37** | 77.30 | **77.71** | 77.56 | 76.69 | 77.62 |
| DeepLabV3 | 72.91 | 71.24 | 3.27 | 3.27 | **9.42** | **72.58** | 71.28 | 37.93 | 6.53 | **72.72** |
| BERT | 83.06 | 71.03 | N/A | 78.65 | **83.06** | 80.31 | 82.61 | **82.80** | 82.81 | 83.02 |
| SalsaNext | 55.80 | 54.22 | 22.24 | 55.79 | **54.92** | 55.52 | **55.67** | 55.12 | 53.08 | 55.02 |
| HRNet | 81.05 | 80.93 | N/A | **80.71** | 0.48 | **81.03** | 81.04 | 80.77 | 80.14 | **81.06** |

Table 3: PTQ Results on all models. Fixed bias: best results over all fixed bias values for each Mantissa / Exponent division. Flex bias: best results over all initialization methods for each Mantissa / Exponent division. Bold numbers mark best results per model for all Fixed bias and Flex bias results, respectively. Bold numbers in the 'Flexible' column indicate results where 'Flexible' PTQ outperforms other PTQ methods.

| Format | CoLA | SST-2 | MRPC | STS-B | QQP | MNLI | QNLI | RTE | Macro avg. |
|---|---|---|---|---|---|---|---|---|---|
| FP32 | 57.27 | 93.12 | 88.36 | 89.09 | 89.72 | 84.91 | 91.58 | 70.40 | 83.06 |
| INT8 | 54.74 | 92.55 | **88.53** | 81.02 | 83.81 | 50.31 | 52.32 | 64.98 | 71.03 |
| 5M2E flex | 58.92 | 91.86 | 87.49 | 88.50 | 89.21 | 81.69 | 80.52 | 64.62 | 80.31 |
| 4M3E flex | 56.51 | 92.32 | 88.16 | **88.99** | **89.79** | 84.87 | **91.69** | 68.59 | 82.61 |
| 3M4E flex | 57.29 | **93.12** | 87.35 | 88.87 | 89.76 | 84.87 | 91.51 | **69.68** | **82.80** |
| 2M5E flex | **60.01** | 92.43 | 88.38 | 88.00 | 89.61 | 84.14 | 91.31 | 68.59 | **82.81** |

Table 4: Per task results for BERT for all GLUE tasks. Best quantized result for each task is marked in boldface.

believe the low performance on fixed format PTQ to be due to the fact that some layers in early the DeepLabV3 backbone have relatively large outliers (e.g. backbone.features.2.conv.0.weight, row 1 column 4), necessitating a relatively large number of exponent bits, while some layers later in the backbone and in the decoder have distributions that require few exponent bits (e.g. backbone.high_level_features.15.conv.3.weight; backbone.high_level_features.16.conv.3.weight; decoder.last_conv.8.weight). This discrepancy is present in other networks, most notably MobileNetV2, however, it is not as prominent as in this network.

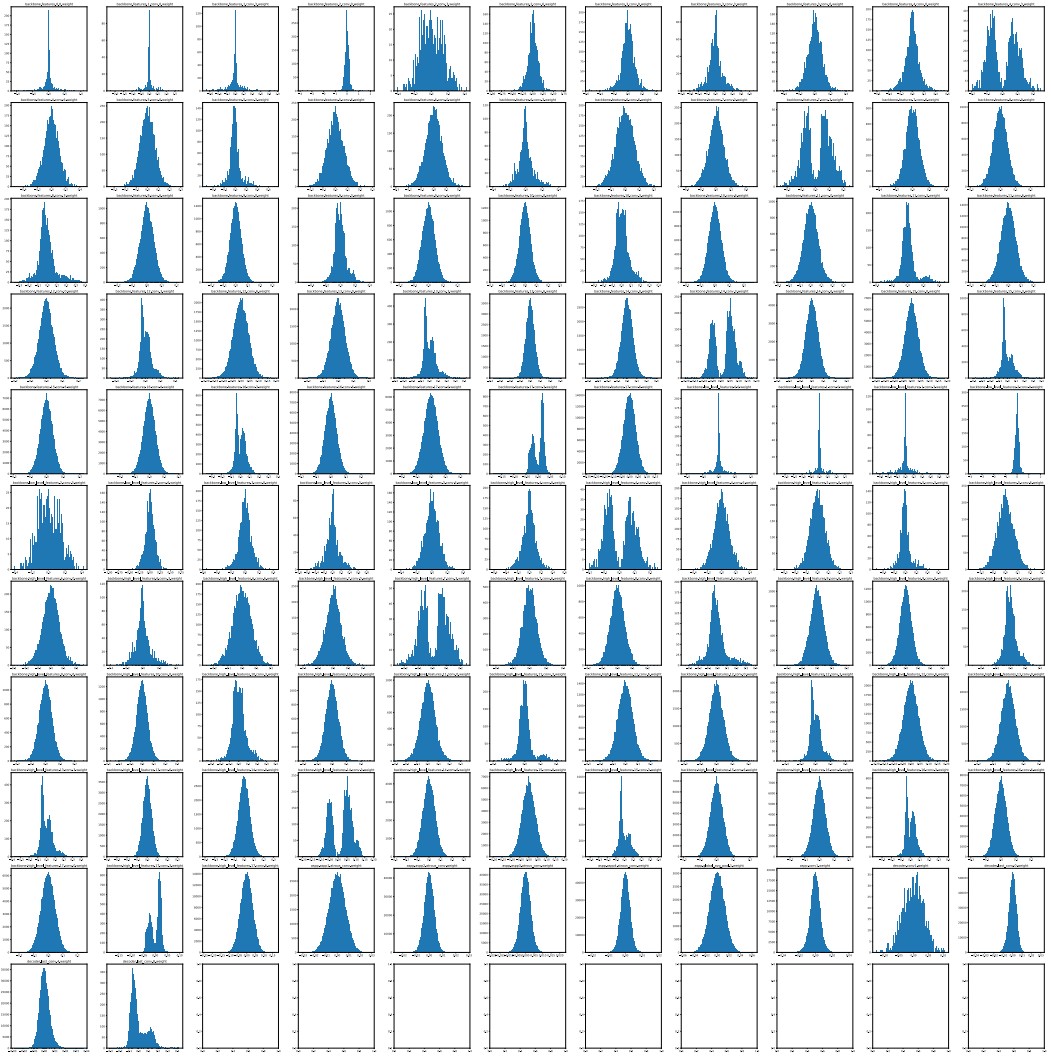

Figure 14: A plot showing the distributions for each weight tensor in the DeepLabV3 model instance used in our experiments.