# OpenReview forum: "FP8 Quantization: The Power of the Exponent"
_NeurIPS.cc/2022/Conference — NeurIPS 2022 Accept_

### Official Review · Reviewer_5HMP · 2022-07-09

**Rating:** 4
**Confidence:** 4
**Soundness:** 3 good
**Presentation:** 2 fair
**Contribution:** 1 poor

**Summary:**

In this paper, the FP8 format quantization algorithm is realized based on a common FP32 deep learning framework for post-training quantization and quantization-aware training. The effects of choosing different bits for the mantissa and exponent on the quantization error and the final accuracy of the model are studied. Experiment results show that FP8 format is better than INT8 format.

**Questions:**

1. What is the major contribution and novelty of this paper? I think the authors should make it more clear. Is it the "chief conclusion" saying that FP8 format has a larger representation range and so has a smaller quantization error on the tensor with outlier? This is a somewhat trivial knowledge. Is it the innovation of this paper to conduct “fast” FP8 quantization simulation in the FP32 framework? If so, the implementation of this paper should be compared with other quantitative simulation frameworks.

2. Line 93, "it has been shown to be indicative of the final loss in a neural network". Can it be confirmed by experiments in this paper?

3. This paper proposes to select the appropriate bits for mantissa and exponent for each layer (best flexible in the experiment? ) Is this off-line selection obtained by traversing the selection of mantissa and exponent bits for the weight and activation of each layer respectively? When selecting the quantization format for each layer, do you aim to minimize the MSE of the tensor, or minimize the scalar product quantization error, or maximize the final model score? If MSE is minimized, will the result with the largest model score be obtained?

4. "best flexible" does not seem to be significantly better than “best flex bias”. The tensor is divided into blocks, and each block adopts different exponent bias, which is equivalent to the integer format of block quantization, also called share exponent. This idea has been put forward in Flexpoint and other work [1][2][3]. Is the block +FP8 scheme used in this paper better than block + INT8? Further consider that the actual deployment requires hardware design. The use of best flex bias or best flexible requires support for various m/e combinations on hardware, while block + INT8 only needs to support one.

5. There should not be a "(PTQ)" under the "INT8" in Table 2.

6. From the incomplete results in Table 2, QAT is not necessary for best flexible? In other words, applying QAT to the best fixed scheme or even INT8 scheme is enough to achieve the optimal accuracy. Is the complex FP8 format such as best flexible necessary?

7. Because the colors in Figure 5 overlap, new colors are generated, which is difficult to distinguish. Maybe change to a line chart?

8. Line 146, "we fit Gaussian distributions in the weights and activations sample, and compute the expected MSE analytically in Fig. 3 (right)" If we are experimenting with the data in the actual model, why should we fit the Gaussian distribution? We should directly use the original data to calculate the quantization error.

9. The MAC results of FP8 tensors are in high-precision (such as FP32). The result after an operator needs to be intercepted as FP8, so, how are the operators divided in various network models? For example, before adding the shortcut path and the main path in ResNet, will they be both quantized as FP8?


**Limitations:**

The scheme in this paper might make the hardware design more complex, because it needs to support different m/e/b, which is mentioned in [3] (see the ref section in "Strengths and Weakness"). I recommend the authors to discuss on the implication for actual hardware implementation.

**Strengths And Weaknesses:**

## Strength

1. This paper did a good work on clearly describing the floating-point format and quantization process.
2. This paper realized the quantization simulation based on the FP32 framework.
3. Experimented with models on various tasks.

## Weakness

I think the major weakness of this work is the lack of novelty and insufficient comparison with baselines. Please see the question section for more detailed discussion.

### Novelty
1. Using floating-point low-bit format in neural network quantization is very common. For example, group-wise quantization [1][2][3], different quantization parameters for different layers [4][5], special schemes for tensor with outliers [6], and QAT scheme for reducing quantization error [7][8] are well discussed in the literature.
2. It seems the experiments do not bring new findings or knowledge beyond existing studies.

### Insufficient comparison
3. There are many other low-bit quantization schemes in the literature (as shown above), this paper should compare with more baselines other than the classical INT8 quantization scheme.

## References
[1] Köster, Urs et al. "Flexpoint: An Adaptive Numerical Format for Efficient Training of Deep Neural Networks." NIPS (2017).

[2] Rouhani, Bita Darvish et al. "Pushing the Limits of Narrow Precision Inferencing at Cloud Scale with Microsoft Floating Point." NeurIPS (2020).

[3] Zhong, Kai et al. "Exploring the Potential of Low-bit Training of Convolutional Neural Networks." IEEE Transactions on Computer-Aided Design of Integrated Circuits and Systems (2022).

[4] Banner, Ron et al. "Post training 4-bit quantization of convolutional networks for rapid-deployment." NeurIPS (2019).

[5] Dong, Zhen et al. "HAWQ: Hessian AWare Quantization of Neural Networks With Mixed-Precision." 2019 IEEE/CVF International Conference on Computer Vision (ICCV) (2019): 293-302.

[6] Zhao, Ritchie et al. "Overwrite Quantization: Opportunistic Outlier Handling for Neural Network Accelerators." ArXiv abs/1910.06909 (2019).

[7] Choi, Jungwook et al. "PACT: Parameterized Clipping Activation for Quantized Neural Networks." ArXiv abs/1805.06085 (2018).

[8] Yamamoto, Kohei. "Learnable Companding Quantization for Accurate Low-bit Neural Networks." 2021 IEEE/CVF Conference on Computer Vision and Pattern Recognition (CVPR) (2021): 5027-5036.

---

> ### Author Response · Authors · 2022-08-01
> **Reply to the comments by Reviewer 5HMP**
>
> The reviewer remarked that the experiments do not bring new findings or knowledge beyond existing studies. We respectfully disagree with the reviewer on this point and would like to refer the reviewer to the ‘Novelty’ section in the general remarks for a discussion on how our findings differ from existing studies.
>
> Regarding the reviewer’s remarks on comparison to other quantization methods: the goal of our paper was to evaluate the performance (PTQ and QAT) of FP8 formats with various levels of flexibility, to the de-facto standard of INT8 quantization, which, coincidentally, is a special case of FP8 quantization. While we agree with the reviewer that an exhaustive comparison of quantization methods is an interesting and valuable goal, we consider this to be outside the scope of our paper. There are signification differences between our method and Flexpoint-like approaches are highlighted in our answer to Q4 below. For our INT8 experiments we use Learned Step Size Quantization [D], which, like PACT, also learns a clipping threshold.
>
> Answers to the reviewer’s specific questions:
> 1. We respectfully disagree with the reviewer that this is trivial knowledge; this is more nuanced than often assumed. We refer the reviewer to the ‘novelty’ section in the general remarks for more details.
>
> 2. We ran a small experiment with injecting a Gaussian noise into the weights of Resnet18 layer, and found the normalized correlation between the MSE in the activations and the final network accuracy is 0.98. We added this experiment to the updated version of the Appendix G. This statement is also in agreement with the literature, e.g. [A,B,C].
>
> 3. The offline selection is based on reconstruction MSE of the input tensor. We can see in Table 1 that, indeed, this approach yields better results for some networks (e.g. MobileNetV2, DeepLabV3, HRNET), but underperforms for others (ViT, SalsaNext). Note that this discrepancy is addressed in the paper (lines 244-247). While scalar product MSE may be a better choice, we chose to use local MSE as a target for two reasons:
>     1. Our analysis in section 2 of the paper shows that local MSE search often yields formats close to those found by scalar product MSE search.
>     2. Scalar product MSE quadratically increases complexity compared to local MSE as we need to consider the quantization parameters of both the input weights and input activations simultaneously.
>
> 4. Note that the "best flexible" format differs significantly from Flexpoint/MSFP.
> With "best flexible", we choose an m/e division for each tensor, and a bias value for each output channel in the weights (and one bias per tensor for the activations), while each individual element stores its own exponent value. For all formats considered in our paper, the 8 bits per tensor element are thus used to store sign, mantissa, and exponent. This differs from Flexpoint/MSFP: in these formats an exponent value (not exponent bias) is shared between blocks of tensor elements, and for each element in a block only sign and mantissa bits are stored. We found the difference w.r.t. these methods large enough to warrant a comparison unnecessary.
> Note that, from a practical perspective, a per-channel bias can be realized similarly to a per-channel scale in INT8, which is available in many HW inference engines.
>
> 5. Thank you, we removed this.
>
> 6. The column headers in table 2 only refer to the method of initializing a network before performing QAT. In case mantissa bits are learned (+m rows in Table 2), best fixed format and best flex bias format turn in to ‘flexible’ formats, as each tensor can learn its own number of mantissa and exponent bits. Likewise, in case a clipping value is learned (+c rows), the ‘best fixed’ format turns into a ‘flexible bias’ format.
> The main conclusion from this table is, similarly to the PTQ results, that a flexible bias is crucial, while flexible mantissa bits marginally improves results.
>
> 7. Thank you for the suggestion, we adapted this.
>
> 8. The analytical results obtained using fitted Gaussian distribution into the experimental data agree with the empirical results obtained using the original data directly (the latter experiment is provided e.g., in Appendix B.2). Fitting a Gaussian curve into the data is aimed at showing that our analytical results are directly applicable to real layers of a network.
>
> 9. In our simulations, we perform output quantization on all operations (linear, conv, addition, concatenation, etc.) except MaxPool, as requantization is not necessary after this step. This includes quantization of the output of all residual blocks and downsample layers, i.e. inputs to the residual additions.
> Furthermore, we assume a Kulisch-like (high precision) accumulator is available for intermediate results of conv and matmul operations. Note that this is standard practice in INT8 hardware.

---

> > ### Comment · Reviewer_5HMP · 2022-08-08
> > **Thank you for the response**
> >
> > Thanks for the response. I think part of A2, A3 and A6 should be added to the text. I still have unsolved concerns and questions:
> >
> > For A4, I know that searching the best FP8 format (different m and e bit widths) for  each layer in this paper is different from Flexpoint or MSFP, but MSFP also has grouping and variable exponent bias (shared exponent is the exponent bias, and E=0 of each independent value). If MSFP can achieve better reasoning results, why do we need the search method? In particular, the mixed format brings a great challenge to the hardware.
> >
> > For innovation and hardware overhead. "Strength of outliers directly correlates with number of expenditure bits required" sounds to be a widely accepted and straightforward view. Because more exponent bits can represent a larger range, and it is the outlier that causes a larger data range. The hardware manufacturers choose 4 or 5 exponent bits because of the universality. If the search algorithm proposed in this paper may select a layer to use 5-bit exponent, the hardware must support 5-bit exponent. Other layers use fewer exponents and more mantissas, which will not save power consumption. For the mixed-precision quantization, without involving the analysis on hardware implementation overhead, we cannot explain the advantages of this work compared with the widely-used INT8 or FP8 format (fixed M and E).
> >
> > For A9，how to deal with BN？

---

> > > ### Author Response · Authors · 2022-08-09
> > > **Reply to the recent comments by Reviewer 5HMP**
> > >
> > > Thank you for the comments. We have added clarifications related to A2, and A3 in the paper, and we will incorporate the improvements related to A6. Thank you for the suggestions.
> > >
> > > In general, our paper does not intend to introduce the flexible FP8 format or claim that it is the best thing to do compared to other methods. As described in the impact/novelty review above, we set out to understand in-depth the benefits of the FP8 format for different exponent bits compared to INT8, while introducing a more principled novel framework that makes this analysis possible.
> > >
> > > We focus on showing the trade-offs from the accuracy side, to make the findings more general. Looping back to your statement, anyone implementing hardware could implement an FP8 format with 5 exponent bits, but it is important for them to realize that a different setting can work significantly better for some models and why. Having 5 exponent bits is not a catch-all solution, if anything, you get between 3 and 4 bits worth of accuracy for uniform distribution with this format, no matter what range. That’s what we describe in this paper. It is up to the user for their specific use-case and hardware to make this trade-off based on these insights. If you make hardware dedicated to Bert inference, you might be good with 4 exponent bits, but for general-purpose model execution, this might be a very poor choice.
> > >
> > > These insights are valuable and hardware agnostic. We agree that having a flexible exponent bit assignment might incur an overhead, more so in dedicated hardware, perhaps less so in general-purpose hardware that already has FP16 implemented; but leave it to the reader to decide this for their specific use-case.
> > >
> > > On A4 – We might have been a bit unclear in our rebuttal. “Best-flex-bias” essentially means we have a learned/set scaling factor per-tensor. Furthermore, we consider per-channel quantization, since it’s a common implementation choice for integer hardware. This means there is a scaling factor per output channel. Since these are common in INT8, we also look at the impact of both on FP8. The MSFP and Flexpoint-like formats can be interpreted as having a scaling factor per-block instead. Analyzing the effect of this is interesting, but quite orthogonal to our findings. One could do a block-wise scaling factor for the FP8 formats as well if deemed interesting. As a matter of fact, a recent work published at ICML 2022 is an approach going in this direction [1]. “Best flexible” is different, however, as this relates to having a different m/e bit setting for each layer in the network, not each block of a tensor. This is far easier to implement in hardware, and is not related to the MSFP scheme.
> > >
> > > On the novelty, we consider the results in the paper are less trivial than is stated. Larger data ranges do not necessarily mean that one format is better than the other. If you have a uniform distribution, no matter over what range, INT8 will be better than the FP8 formats. This even occurs if you have wildly large ranges because the range is estimated for the quantization grid and taken into account with the scaling factor. This finding also holds for other distributions like the Gaussian and student-t we show in the paper, where the trade-offs change and more exponent bits are required for the best representation. Specifically, we show that it is the skewness of the distribution, namely outliers that are far off from where most values lie, that dictate if exponent bits are helpful or harmful. Not the range itself.
> > >
> > > Regarding BN. We follow the approach of [2] which allows us to re-estimate batch norm statistics during training. Afterwards, we absorb the batch norm scaling into the per-channel quantization scale, thus during the forward pass, there is no extra computation for BN. Having a per-channel quantization scale is a standard practice of HW-realistic quantization simulation [3,4].
> > >
> > > References
> > >
> > > [1] Yeh, Thomas, et al. "Be Like Water: Adaptive Floating Point for Machine Learning.", ICML 2022
> > >
> > > [2] Nagel, Markus, et al. "Overcoming Oscillations in Quantization-Aware Training.", ICML 2022
> > >
> > > [3] Krishnamoorthi, Raghuraman. "Quantizing deep convolutional networks for efficient inference: A whitepaper."
> > >
> > > [4] Nagel, Markus, et al. "A white paper on neural network quantization." arXiv preprint (2021).

---

### Official Review · Reviewer_iumQ · 2022-07-09

**Rating:** 8
**Confidence:** 4
**Soundness:** 4 excellent
**Presentation:** 3 good
**Contribution:** 4 excellent

**Summary:**

FP8 is discussed both in the context of quantization-aware training and in the context of post-training quantization.  A very nice analysis of the expected quantization error is provided by the authors when the input to the quantizer is normal/ uniform/student's distribution. One of the most important and striking conclusions is that FP8 is a better format for inference than INT8, even though the latter has been the standard format up to now. In my opinion, this is a critical conclusion since switching between floating-point formats (used for training) and integer formats (used for inference) is a pain.



**Questions:**

1. Bert-base opts for 3M4E in table 1. This may be due to heavy-tailed distributions of weights and activations. Could you please estimate the type of distribution Bert has in the FWD pass? Could a plot be included?
2. What task does Bert base use here (word masking)? I'm not familiar with the baseline accuracy of  "83.06" in table 1 (baseline accuracy should be ~86).

XXXXX
Thank you for your detailed response. Could you please share the code?




**Limitations:**

The current work has no negative societal impact.

**Strengths And Weaknesses:**

Strengths:
1. The paper provides a rigourous approach to proving that FP8 outperforms INT8 in inference. 5M2E is found to minimize MSE of normal distributions, and thus is of particular interest. This is a well-written, novel paper with clear practical implications.
2. The results of the analysis and simulation are in close agreement.

Weakness:
1. I do not find the FP simulator to be novel or interesting.
2. It was a great work by the authors to provide MSE based on clipping value (set via exponent bias). Next, they should find the derivative, set it to zero, and find the roots or, if that is not possible, numerically solve it. Otherwise, analytical expressions have less practical value.

---

> ### Author Response · Authors · 2022-08-01
> **Reply to the comments by Reviewer iumQ**
>
> “I do not find the FP simulator to be novel or interesting.”
> * We would like to refer the reviewer to the ‘novelty’ section in the general comments, where we address this concern.
>
> “It was a great work by the authors to provide MSE based on clipping value (set via exponent bias). Next, they should find the derivative, set it to zero, and find the roots or, if that is not possible, numerically solve it. Otherwise, analytical expressions have less practical value.”
> * In section 3, the clipping ranges are obtained using numerical minimization. Unfortunately for Gaussian and Student’s-T distributions there is no analytical expression for the optimal clipping range, and finding the roots must be done numerically. For example, for a Gaussian distribution, there is a non-trivial combination of logarithmic and polynomial functions which does not allow us to re-express the clipping range analytically. Would that derivation be of an interest for the reviewer, we can provide it in a follow up discussion.  Although this leads to certain loss of elegancy in the analysis, we have no concerns about the quality of the numerical solutions. In all the cases the functions minimized have one clear global minimum, and there are no convergence issues.
>
> “Bert-base opts for 3M4E in table 1. This may be due to heavy-tailed distributions of weights and activations. Could you please estimate the type of distribution Bert has in the FWD pass? Could a plot be included?”
> * The distributions observed in the forward pass of Bert are generally Gaussian. However, as it was observed in [E] there are strong outliers in the tails. We provided some plots for this in the updated version of the supplementary. Additionally, we would like to refer the reviewer to Figure 11 in the supplementary material, which shows the per-layer SQNR for the full BERT network.
>
> “What task does Bert base use here (word masking)? I'm not familiar with the baseline accuracy of "83.06" in table 1 (baseline accuracy should be ~86).”
> * The reported scores are the GLUE macro average on a validation split of the GLUE benchmark task. This result is generated using the BERT-base uncased model published on huggingface.com. Note that these results are not test set scores. Other published works have reported similar baseline scores (e.g., [E]).

---

> > ### Comment · Reviewer_iumQ · 2022-08-10
> > **Code**
> >
> > Thank you for your detailed response. Could you please share the code?

---

> > > ### Author Response · Authors · 2022-08-10
> > > **Code**
> > >
> > > We will release the source code, including the expected MSE computation, FP8 quantization simulation, and the code for finding the best bias and best flexible formats. Due to legal reasons out of our control, the code has not been released yet. We will be able to provide the code by the camera-ready version. Hopefully that is sufficient :)

---

### Official Review · Reviewer_5G84 · 2022-07-10

**Rating:** 5
**Confidence:** 3
**Soundness:** 3 good
**Presentation:** 3 good
**Contribution:** 3 good

**Summary:**

This paper first analyzes the effect of the number of bits for the mantissa and exponent in FP8 data format, and then proposes a method to learn them with gradient descent. Empirical results on various tasks with various models show that the proposed FP8 quantization has a clear performance improvement for post-training quantization but performs similarly or even slightly worse than previously commonly used uniform INT8 quantization.

**Questions:**

Please address the questions above.

**Ethics Review Area:**

["I don’t know"]

**Limitations:**

I did not find extra sections discussing about the limitations or broader impact, but some are talked about in the related work and conclusion.
It would be better if the authors can discussion about them in more detail.

**Strengths And Weaknesses:**

The paper is overall well written and easy to follow. The discussions on the effect of using different #bits for the mantissa and exponent to the quantization error are also intuitive. Extensive experiments on various tasks and various domains are performed to validate the efficacy
of the proposed method.

Still some parts of the method and experiment sections need some more clarifications.
- Lines 119-120: Why the MSE can be approximated by (7). Are there any intuitions, explanations, or references?
- Table1: most of the time, "Fixed" performs similarly to "flex bias" and "flexible" on all models except DeepLabV3. Are there any explanations for this observation?

---

> ### Author Response · Authors · 2022-08-01
> **Reply to the comments by Reviewer 5G84**
>
> “Lines 119-120: Why the MSE can be approximated by (7). Are there any intuitions, explanations, or references?”
> * The derivation is provided in appendix A.2. The intuition behind this approximation is that the cross-terms in (18) are negligible compared to the first two square terms. And those two terms are equal to the rounding errors for both inputs.
>
> “Table1: most of the time, "Fixed" performs similarly to "flex bias" and "flexible" on all models except DeepLabV3. Are there any explanations for this observation?”
> * Note that there are significant differences for ResNet18 and MobileNetV2 as well. A degradation of >1% in performance is generally considered significant.
> The larger difference for DeepLabV3 is likely caused by a strange mix of tensor distributions: Some weights in the MobileNetV2 backbone model used in these experiments (specifically those early in the network) have very large outliers, requiring many exponent bits. However, some weights in the segmentation head have weight distributions which require very few exponent bits, and where large exponent bit widths harm. We have extended this analysis in the supplementary work in the updated version, a plot of weight distributions has been included.
>
> We thank the reviewer for suggesting a limitation/impact section. We agree that this can help to clarify the scope of our paper and will add such a section in the camera-ready version as we mentioned in the general remarks.

---

### Official Review · Reviewer_tuZq · 2022-07-11

**Rating:** 4
**Confidence:** 4
**Soundness:** 2 fair
**Presentation:** 3 good
**Contribution:** 3 good

**Summary:**

This paper investigates how the choices that can be made to leverage the FP8 format (i.e. including the important choice of the number of bits for the mantissa and exponent), and these investigations deliver analytical results on which settings for better performance. Following their analytic contributions, this work also shows how these findings translate to real networks, provide an efficient implementation for FP8 simulation, and a new algorithm that enables the learning of both the scale parameters and number of exponent bits in the FP8 format. The main conclusion from this work is that FP8 format is more suitable than INT8 in terms of the accuracy, when doing post-training quantization for a wide range of networks, the FP8 format is better than INT8 in terms of accuracy. Also, this work delivers a recommendation that the choice of the number of exponent bits is driven by the severity of outliers in the network (under FP8 format). A sensitivity study on quantization-aware training is also carried out.

**Questions:**

- Can your method benefit the quantization-aware training in terms of the efficiency? If yes, how?
- Can you comment on the potential impacts on the inference performance, given the solution and your recommendations (i.e. 5M2E & 4M3E)?

**Limitations:**

- The comparison is performed between FP8 and INT8 fake quantization, and I suggest the authors to make it clear upfront.

- The accuracy is expected to be good by using FP8 format, since it can have a wider coverage than INT8. However, this can bring huge overheads in terms of the inference performance (see next), and it shall not be overlooked.

- The inference performance is hard to benefit from the concluded 5M2E and 4M3E: Minifloats are commonly used in embedded devices, which need to be emulated in software. These overheads are extra burdens to the inference over INT8 fake quantization.


**Strengths And Weaknesses:**

Strengths:
- A sensible insight and the solution is convincing (for the accuracy)
- A method to make simulated quantization fast
- Evaluation results are sufficient to back up the current claim

Weaknesses:
- Limited insights (i.e., this work essentially describes a usage of minifloats for DNNs)
- The proposed method is only tailored to an efficient quantization procedure
- No discussions on the potential impacts on the inference performance
- Low reproducibility (No source codes available)

---

> ### Author Response · Authors · 2022-08-01
> **Reply to the comments by Reviewer tuZq**
>
> The reviewer noted that our paper provided limited insights. We would like to emphasize that the insights provided by this paper are potentially impactful, given broader industry developments, as mentioned in the general comments.
>
> “The proposed method is only tailored to an efficient quantization procedure”
> * We are unsure how to interpret this weakness. Could the reviewer elaborate on this remark?
>
> The reviewer mentioned several concerns regarding the efficiency of our method. In addition to our remarks on hardware efficiency in the general comments, we would like to point out that our method allows more efficient (simulated) quantization-aware training through a more efficient FP8 quantization simulator. In this respect our method benefits quantization-aware training in terms of efficiency, however, we do not optimize our networks directly for efficiency, as stated in the general comments.
> Furthermore, our paper only concerns simulated quantization. We consider fully quantized training or on-device quantization aware training outside of the scope of this paper and reserve this for future work.
>
>
> “The comparison is performed between FP8 and INT8 fake quantization, and I suggest the authors to make it clear upfront.”
> * We are sorry that this was not clear from the paper. We would like to note that simulated quantization is common practice in INT8 quantization research (e.g. [A,B,D,E,F]), in terms of accuracy, simulated quantization correlates well with on-device accuracy of a quantized model.
>
> Lastly, the reviewer remarked on reproducibility. To increase reproducibility, we will release source code, including the code for the expected MSE computation, the code for our FP8 quantization simulation, and the code for finding the best bias and best flexible formats.

---

> > ### Comment · Reviewer_tuZq · 2022-08-10
> > **Please address the following issues.**
> >
> > Correction: "The proposed method is only tailored to an efficient quantization procedure, in terms of the accuracy only".
> >
> > I have read the response (including general comments and the individual reply), as well as other reviewers' comments. I am still concerned about the efficiency issue, and I agree with Reviewer 5HMP on the insufficient comparison. See below.
> >
> > 1. If you assume your method allows more efficient QAT, please provide the empirical evidence quantitatively to support the claim.
> >
> > 2. Comparisons between INT8 and FP8 shall be considered essential to jusitfy the efficiency challenges. The bottleneck lies on the available number of FP registers and respective BMI extenions. As far as I know, there are no such supports in current products.
> >
> > 3. I acknowledge that I am aware of simulated quantization is common practice in INT8 quantization research, while I wrote the initial review. This is also important to restrict the scope of your contribution while other lines of works appear (such as [1]).
> >
> > 4. Following the above point, how would your approach compare with [1] quantitatively and qualitatively?
> >
> > [1] Zhewei Yao, et al. HAWQ-V3: Dyadic Neural Network Quantization. ICML'21.

---

> > > ### Author Response · Authors · 2022-08-10
> > > **Reply to the comments of Reviewer tuZq**
> > >
> > > 1. We benchmarked the efficient FP8 quantizer from section 5 on a GPU. It gives substantial speed up compared to a naïve approach (element-wise computation of the exponent/mantissa without considering the floating point grid as a union of m uniform grids). For example, the combined forward+backward pass of the quantization operation on 1000 batches of size 64x512x7x7 takes 1.20 ± 0.09 sec for our approach versus 2.33 ± 0.01 sec for the naïve approach (averaged over 10 runs).
> > >
> > > 2. As we stated in the general comments, we intentionally leave the hardware implementation aspects out of the scope of our paper and focus on showing the trade-offs from the accuracy side, to make the findings more general. These insights are valuable for hardware design and are hardware agnostic. Even more so now that there is a standardization on the way for FP8 formats for AI workloads (https://bit.ly/3bMJEtx). Any hardware consideration is only specific to the designers intended use-case.  We wonder if the reviewer could elaborate on what parts of the general comments specifically they do not agree with, since we’re very open to discussion on this point.
> > >
> > > 3. Thank you for the suggestion, we will incorporate this into the paper. We agree that it is important to restrict the scope. As stated in the general reply we will add limitations and broader impact section which will also clarify the scope.
> > >
> > > 4. The work [1] is a mixed-precision uniform quantization method. Using a different bit-width for each layer for INT/FP quantization is orthogonal to our method and the two approaches can be combined. You could have for example some layers in an FP6 format and others in an FP5 format if you wanted, or even combine FP formats with INT formats. For this reason, no direct comparison is possible between our work and [1], and the two methods are entirely complementary.

---

> > > > ### Comment · Reviewer_tuZq · 2022-08-10
> > > > **Please address the comments**
> > > >
> > > > 1. I assume the original paper (as well as the appendix) does not provide this part. Can you give out the comparison comprehensively (against FP32/FP8/INT8) in terms of the performance?
> > > >
> > > > 2. I already identify the issue in a previous reply. As I identified, I consider the comparison between INT8 and FP8 is essential (and my previous comments already elaborate why). Also, your general comments are not convincing enough to answer my previous comments as well.
> > > >
> > > > 3. Thanks. Please restrict the scope of your work properly.
> > > >
> > > > 4. The core spirits of HAWQ-V3 is not as claimed by the authors. The work introduces dyadic scaling into the quantization, so that the inference procedure donot require extra lookup (which can be highly expensive in term of the computation costs) but only uses a few bit-level operations (e.g. shifting). Therefore, this is the reason why I keep emphasizing the comparison with INT8 is important.
> > > >
> > > > Also, I do not think it's as easy as the authors describe for the combination between your approach and theirs. Adapting FP8 formats can add a considerable amount of challenges for bit-level operations (since it's very hard to extract and exploit these bit-level patterns).
> > > >
> > > > I suggest the authors addressing the concerns in a more specific manner. My major concern still lies on PTQ performance under FP8 against INT8 ones, and how previous lessons on INT8 can be beneficial to FP8 ones. Without a sufficient amount of empirical evidence, I won't adjust my score for a positive lift of this paper.

---

### Author Response · Authors · 2022-08-01
**General introduction and remarks on Impact, Novelty, and Hardware impliciations**

We thank all reviewers for their thoughtful comments and useful feedback. Some general comments were made about the impact of the work, the novelty and implications for hardware. We address these comments in this section. We will add limitations and broader impact sections to the  camera-ready version.

For reviewer-specific comments, we replied to individual reviews.

## Impact
We compare FP8 to INT8 as these two formats are commonly discussed as standards for efficient deep learning inference. FP8 is becoming widespread as can be seen here: https://bit.ly/3Sd8wey, with multiple vendors like Nvidia, Intel, Graphcore, AMD, IBM chips etc. all moving to support FP8. For this reason, the comparison between these two formats is significant and impactful. We have extended our QAT results to complete the MobileNetV2 results and included results on BERT as well.

This is less so the case for other methods the reviewers mentioned. However, our analysis of the performance of INT8/FP8 on common distributions, per-layer and full network, can also be done for these or other methods. This is beyond the scope of our paper but would be interesting future work.

## Novelty
To our knowledge, we are the first paper to systematically analyze the impact FP8 of quantization, i.e., adding exponent bits, compared to commonly used INT8 quantization. The novelty is mainly in the procedure where we analyze this on various levels (distribution, layer, network), and show that the insights from the distribution level translate to the network level. Our main insight – strength of outliers directly correlates with number of exponent bits required – we have not seen in any other paper. Hence, we don’t regard it as trivial. In fact, discussing this point with others in the quantization community, many are surprised by the results. We see that e.g. Graphcore, Tesla (https://bit.ly/3PGV3Kg) and Nvidia implement FP8 formats with 4 and 5 exponent bits, although our analysis shows that these formats are rarely optimal, and 2 and 3 bits would be preferable for inference. Insights from this paper could influence other vendors’ decisions in this respect.

Furthermore, we introduce an efficient way to simulate FP8 quantization in FP32 hardware. This formulation allows for fast experimentation (both in post-training and QAT settings) and is extended to allow learning the mantissa/exponent bit split and any (FP32) bias value. This approach differs from previous methods of simulating as it is faster than e.g. lookup table based simulations, and does not require specific hardware (unlike [F]).

## Hardware numbers/comparison
The hardware impact of doing INT8 or FP8 is difficult to assess, and very sensitive to specific design choices. We try to avoid these specific choices and implementations and focus on generally applicable knowledge in our paper.

Assessing the difference between INT8 and FP8 impact on hardware and efficiency is not trivial. For networks that are data transfer bandwidth limited (e.g. ResNet last layers with many weights per activation; image-to-image networks on HD inputs), data transfer dominates power and latency, and both formats can just be regarded as two different 8-bit formats with similar effects on power and latency.

For calculations themselves, one could argue that FP8 uses slightly larger die space, and uses more power for additions and multiplications [G]. However, this can change significantly depending on how extra logic is built into the circuit, e.g., logic for overflow checks, NaN handling, multiplier and accumulater design choices, sparsity support, are parts of the MAC unit turned off with repeating 0s, etc. On top of that, many such formats would be implemented in multi-purpose designs, that support not only FP8, but e.g., also FP16/32, INT8/16. In such designs, the difference in compute latency and power for INT8 and FP8 generally disappears.

For this reason, we believe that a proper comparison between formats with equal bits is warranted. For teams that want to design their own hardware, they can take our accuracy analysis into account, and make their own trade-offs on the hardware side based on their own specific use-case and design.

### References
[A] Zhang, X., et al. "Accelerating very deep convolutional networks for classification and detection." TPAMI 2015

[B] Nagel, M., et al. "Up or down? adaptive rounding for post-training quantization." ICML 2020.

[C] Moons, B., et al. "Distilling optimal neural networks: Rapid search in diverse spaces." CVPR 2021.

[D] Esser, S. K., et al. “Learned Step Size Quantization.” ICLR 2020.

[E] Bondarenko, Y., et al. “Understanding and Overcoming the Challenges of Efficient Transformer Quantization.” EMNLP 2021

[F] Köster, U. et al. "Flexpoint: An Adaptive Numerical Format for Efficient Training of Deep Neural Networks." NIPS 2017

[G] Horowitz, M., "Computing's energy problem", ISSCC 2014, pp. 10-14

---

### Meta-Review · Area_Chair_yMKW · 2022-08-23

**Recommendation:** Accept
**Confidence:** Certain

**Metareview:**

This paper had mixed reviews.

One very positive expert reviewer (8) pointed out this paper used a rigorous approach to showing than FP8 can outperform INT8 in inference, which I agree is very interesting and useful.

 Another reviewer gave borderline acceptance (5), and I did not find any remaining concerns following the authors' response.

One reviewer gave borderline reject (4), but I did not find any remaining coherent major concerns following the authors' rebuttal. Also, this reviewer seemed less experienced, so I down-weighted this reviewer's score.

Another reviewer gave borderline reject (4) with the following remaining concerns:

(1) "I think this scheme does not show advantages over the existing work. "
But I don't think this is true, since as far as I know previous work did not show such an advantage of FP over INT.

(2) "The practical application of the algorithm in this paper will bring extra overhead. "
I agree the authors should give more details here (especially for flexible), but at least for the flex bias method, the extra overhead seems quite reasonable (as this is similar to the standard method used for INT), so I'm not sure what is the issue.

(3) "it is a very intuitive view that we should adopt a format with more exponent bits on the data with a large distribution range".
But the authors' response correctly said this is not true (as a uniform distribution would be better represented using INT, no matter what it's range), and is not what they are saying.

Therefore, I think the reviewer had some errors in understanding here, and so I down-weighted this reviewer's score.

Also, the following paper seems relevant:
A Block Minifloat Representation for Training Deep Neural Networks, ICLR 2022

**Award:**

No

---

### Decision · Program_Chairs · 2022-09-14

Accept